# SARS-CoV-2 N protein promotes NLRP3 inflammasome activation to induce hyperinflammation

Pan Pan [1,2,6], Miaomiao Shen [3,6], Zhenyang Yu [3,6], Weiwei Ge [3], Keli Chen [3], Mingfu Tian [2,3], Feng Xiao [3], Zhenwei Wang [2], Jun Wang [4], Yaling Jia [2], Wenbiao Wang [2], Pin Wan [2], Jing Zhang [2], Weijie Chen [2], Zhiwei Lei [2], Xin Chen [2], Zhen Luo [2,5], Qiwei Zhang [2,5], Meng Xu [1], Geng Li [2,5✉], Yongkui Li [2,5✉] & Jianguo Wu [1,2,3,5✉]

Excessive inflammatory responses induced upon SARS-CoV-2 infection are associated with severe symptoms of COVID-19. Inflammasomes activated in response to SARS-CoV-2 infection are also associated with COVID-19 severity. Here, we show a distinct mechanism by which SARS-CoV-2 N protein promotes NLRP3 inflammasome activation to induce hyper-inflammation. N protein facilitates maturation of proinflammatory cytokines and induces proinflammatory responses in cultured cells and mice. Mechanistically, N protein interacts directly with NLRP3 protein, promotes the binding of NLRP3 with ASC, and facilitates NLRP3 inflammasome assembly. More importantly, N protein aggravates lung injury, accelerates death in sepsis and acute inflammation mouse models, and promotes IL-1β and IL-6 activation in mice. Notably, N-induced lung injury and cytokine production are blocked by MCC950 (a specific inhibitor of NLRP3) and Ac-YVAD-cmk (an inhibitor of caspase-1). Therefore, this study reveals a distinct mechanism by which SARS-CoV-2 N protein promotes NLRP3 inflammasome activation and induces excessive inflammatory responses.

[1] The First Affiliated Hospital of Jinan University, Guangzhou, China. [2] Guangdong Provincial Key Laboratory of Virology, Institute of Medical Microbiology, Jinan University, Guangzhou, China. [3] State Key Laboratory of Virology, College of Life Sciences, Wuhan University, Wuhan, China. [4] The Affiliated ShunDe Hospital of Jinan University, Foshan, China. [5] Foshan Institute of Medical Microbiology, Foshan, China. [6] These authors contributed equally: Pan Pan, Miaomiao Shen, Zhenyang Yu. ✉email: lg@gzucm.edu.cn; lyk070@jnu.edu.cn; jwu898@jnu.edu.cn

Severe acute respiratory syndrome coronavirus 2 (SARS-CoV-2), so named by the International Committee on Taxonomy of Viruses, and the consequent severe coronavirus disease 2019 (COVID-19) have now become the most serious public health problem worldwide[1,2]. The virus mainly infects the respiratory tract of humans, causing fever, dry cough, fatigue, shortness of breath, body aches, diarrhea, and other symptoms[3,4]. A small number of critically ill patients may progress to acute respiratory distress syndrome (ARDS), metabolic acidosis, septic shock, and clotting dysfunction, or even death in severe cases[5,6]. The infection and transmission of SARS-CoV-2 has caused great harm to human health and social development. Therefore, the pathogenesis of SARS-CoV-2 is an urgent scientific problem to be solved.

SARS-CoV-2 infects human respiratory system and causes severe inflammatory responses[7,8]. Evidence shows that a dysregulated innate immune response contributes to the clinical presentation of patients with severe COVID-19[9]. Inflammasomes are an important part of the innate immune system that can recognize cellular stresses and infections[10,11]. Inflammasomes are named according to different sensing proteins and there are four main types as follows: NLRP1, NLRP3, NLRC4, and AIM2[12]. Among them, the NLRP3 inflammasome has important functions in RNA virus infection[13,14]. NLRP3 inflammasome consists of a sensor protein (NLR family PYRIN domain containing-3, NLRP3), an adaptor protein (apoptosis-associated speck-like protein containing a caspase recruitment domain, ASC), and an effector protein (Caspase-1)[15]. NLRP3 protein contains three domains: Pyrin domain (PYD), Nucleotide-binding domain, and Leucine-rich repeat domain[14]. During activation of the NLRP3 inflammasome, NLRP3 PYD interacts with ASC PYD, promotes ASC oligomer formation, and provides a platform for caspase-1 activation. Active caspase-1 is formed by autocatalytic cleavage, which then catalyzes proteolytic processing of pro-interleukin (IL)-1β into mature IL-1β[16]. Excessive IL-1β stimulates systemic inflammation responses by activating various signaling pathways, such as nuclear factor-κB (NF-κB) and c-Jun N-terminal kinase pathways[17,18]. As a result, large amounts of cytokines are released, including IL-6, tumor necrosis factor (TNF), interferon (IFN)-α, IFN-β, and transforming growth factor-β[18], which can lead to a "cytokine storm" in acute inflammatory diseases. Studies have reported that inflammasomes are induced upon SARS-CoV-2 infection and are associated with COVID-19 severity[19,20].

SARS-CoV-2 belongs to the β-genus Coronavirus in the Coronaviridae family[21]. The spherical viral particle encapsulates its genome, an unsegmented, positive-sense single-stranded RNA with a size of ~30 kb[21]. The viral genome is enclosed by a nucleocapsid (N) protein, which is buried inside phospholipid bilayers and is covered by a spike (S) protein. The membrane (M) protein and the envelope (E) protein are located among the S proteins in the virus envelope. In addition, the virus encodes a series of accessory proteins (ORF3a, ORF6, ORF7a, ORF7b, ORF8, and ORF10)[21,22]. Previous studies have found that SARS-CoV, which also belongs to the coronavirus subfamily, can activate NLRP3 inflammasome[23,24]. Further studies have revealed that E and ORF3a of SARS-CoV activate NLRP3 inflammasome by changing the K$^+$ ion permeability of plasma membrane and the production of mitochondrial reactive oxygen species[23,24]. The specific molecular mechanism by which SARS-CoV-2 activates NLRP3 inflammasomes is unclear.

Here we show that SARS-CoV-2 N protein induces proinflammatory cytokines through promoting the assembly and activation of the NLRP3 inflammasome. The N protein aggravates lung injury and accelerates death in acute inflammation mouse models through facilitating the NLRP3 inflammasome activation. We also show that SARS-CoV-2 N protein promotes the assembly of the NLRP3 inflammasome through direct interaction with NLRP3 protein. Moreover, SARS-CoV-2 N-induced lung injury and cytokine production are blocked by MCC950 (a specific inhibitor of the NLRP3) and Ac-YVAD-cmk (an inhibitor of the caspase-1). Compared with $NLRP3^{+/+}$ mice, N-induced lung injury and cytokine production are also suppressed in $NLRP3^{-/-}$ mice. This work thereby identifies a molecular mechanism by which SARS-CoV-2 infection causes ARDS, provides evidence that links the NLRP3 inflammasome and lung injury, and suggests that MCC950 and Ac-YVAD-cmk might function as potential therapeutic agents for the prevention and treatment of COVID-19.

## Results

**SARS-CoV-2 N protein induces proinflammatory responses in cells and mice.** SARS-CoV-2 infection induces hyperinflammatory syndromes characterized by overexpression of proinflammatory factors[5,25,26]. Here we initially analyzed and compared the expression statuses of cytokines and chemokines in mock-infected and SARS-CoV-2-infected macrophages and dendritic cells (DCs) based on the data set GSE155106 from the Gene Expression Omnibus (GEO) database. The data showed that expression of proinflammatory cytokines (IL-1β, IL-6, TNF, IL-11, IL-10, and IL-27) and chemokines (C-X-C motif chemokine ligand 8 (CXCL8), C–C motif chemokine ligand 2 (CCL2), CXCL10, and IL-7) were significantly induced upon SARS-CoV-2 infection in macrophages (Fig. 1a) and DCs (Fig. 1b). As inflammasome-mediated mature IL-1β induces the release of cytoplasmic proteins and factors, and plays an important role in initiating "cytokine storm"[18], we evaluated the effect of SARS-CoV-2 proteins on the production of mature IL-1β. SARS-CoV-2 genes (N, M, E, 3a, 6, 7a, 8, and 10) encoding for three structural proteins (N, M, and E) and five accessory proteins (ORF3a, ORF6, ORF7a, ORF8, and ORF10) were synthesized based on the sequences of SARS-CoV-2 (GenBank accession number MN908947.3). The synthesized genes were cloned to pcDNA3.1(+) expression vector with HA-tag or Flag-tag. Human monocytic (THP-1) cells and human embryonic kidney 293T (HEK293T) cells were then transfected with these plasmids for 48 h. Quantitative reverse transcriptase-PCR (qRT-PCR) data showed that the mRNAs of SARS-CoV-2 genes N, 3a, 7a, 8, and 10 were expressed at very high levels in THP-1 cells (Supplementary Fig. 1a) and HEK293T cells (Supplementary Fig. 1b). Western blotting results showed that SARS-CoV-2 proteins N, M, E, ORF3a, ORF6, ORF7a, and ORF8, except protein ORF10, were expressed in transfected HEK293T cells (Fig. 1c). We noticed that the N protein was expressed at the highest level compared with other proteins (Fig. 1c).

The role of SARS-CoV-2 proteins in regulating the secretion of mature IL-1β were assessed in a reconstructed NLRP3 inflammasome system as described previously[14]. HEK293T cells were co-transfected with plasmids encoding the three components (NLRP3, ASC, and pro-Caspase-1) of the NLRP3 inflammasome and the substrate pro-IL-1β to generate a NLRP3 inflammasome system. The HEK293T-NLRP3 inflammasome system cells were subsequently transfected with plasmids encoding N, M, E, 3a, 6, 7a, 8, and 10 proteins, respectively. Enzyme-linked immunosorbent assay (ELISA) results showed that IL-1β secretion was significantly induced by N protein in this system (Fig. 1d). In addition, phorbol-12-myristate-13-acetate (PMA)-differentiated THP-1 macrophages were transfected with plasmids encoding N, M, E, 3a, 6, 7a, 8, and 10 proteins, and stimulated with Nigericin (a specific activator for the NLRP3 inflammasome). We noted that the secretion of mature IL-1β was significantly induced by Nigericin and the Nigericin-induced mature IL-1β secretion was further remarkably enhanced by the N protein (Fig. 1e). Collectively, these results demonstrated that SARS-CoV-2 N

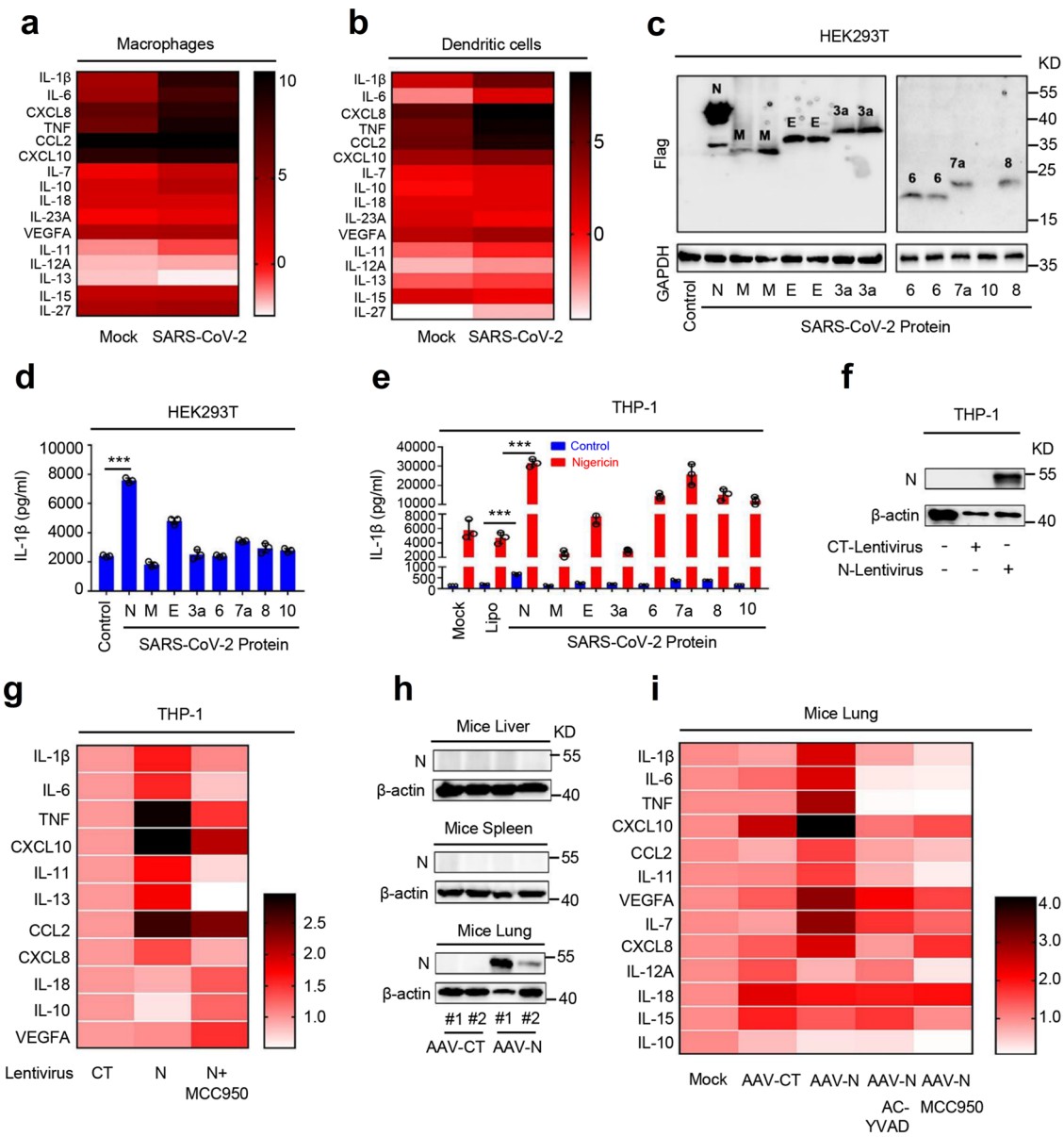

**Fig. 1 SARS-CoV-2 N protein induces inflammatory responses.** Transcriptional level of indicated gene in macrophage (**a**) and dendritic cells (**b**) infected with SARS-CoV-2 (GSE155106, RNA-Seq data from GEO database) were analyzed and the values were showed in logarithmic form. **c** HEK293T cells were transfected with plasmids encoding N, M, E, 3a, 6, 7a, 8, and 10 proteins for 24 h. The indicated proteins in cell extract were analyzed by WB. **d** HEK293T cells were co-transfected with plasmids encoding NLRP3, ASC, pro-Caspase-1, or pro-IL-1β, and transfected with plasmids encoding N, M, E, 3a, 6, 7a, 8, and 10 for 48 h. Supernatants were analyzed by ELISA for IL-1β. **e** PMA-differentiated THP-1 macrophages were transfected with plasmids encoding N, M, E, 3a, 6, 7a, 8, and 10 proteins for 48 h, and then stimulated with 2 μM Nigericin or DMSO for 2 h. IL-1β in cell supernatants was measured by ELISA. **f**, **g** THP-1 cells were stably infected with Lentivirus-CT or Lentivirus-N, differentiated into macrophages. Lentivirus-N macrophages were then treated with MCC950 (0.01 μM) for 1 h. Cell lysates were analyzed by immunoblotting (**f**). The indicated gene mRNA was quantified by qRT-PCR (**g**). **h**, **i** C57BL/6 genetic background mice were tail vein-injected with 300 μl containing 5 × 10^11 vg of AAV-Lung-EGFP (n = 8) or AAV-Lung-N (n = 16); after 2 weeks, they were treated with MCC950 (50 mg/kg) or Ac-YVAD-cmk (8 mg/kg) by intraperitoneal injection for AAV-Lung-N (n = 8) mice; after 3 weeks, two mice for each group were killed and the lungs were collected. The indicated proteins were analyzed by WB (**h**). The indicated gene mRNAs were quantified by qRT-PCR (**i**). Mock means healthy donors (**a**, **b**), untreated cells (**e**), and injection of the same dose of PBS (**i**). Control means transfected with empty plasmids (**c**, **d**). Lipo means transfection with reagents Lipo2000 (**e**). Data are representative of three independent experiments and one representative is shown. Error bars indicate SD of technical triplicates. Values are mean ± SEM. *P ≤ 0.05, **P ≤ 0.01, ***P ≤ 0.001, two-tailed Student's t-test. Source data are provided as a Source Data file.

protein plays an important role in the induction of mature IL-1β secretion.

Next, the effect of N protein on the regulation of proinflammatory cytokines and chemokines was assessed in PMA-differentiated THP-1 macrophages. THP-1 cells were infected with N-Lentivirus (Lentivirus carrying the N gene) and CT-Lentivirus (as a negative control) to generate two cell lines stably expressing N-Lentivirus and CT-Lentivirus, respectively (Fig. 1f). The stable cells were subsequently differentiated into macrophages. qRT-PCR analyses showed that the expression of *IL-1β, IL-6, TNF, CXCL10, IL-11, IL-13, CCL2,* and *CXCL8* mRNAs was notably elevated by Lentivirus-N, whereas the expression of *IL-18,*

*IL-10*, and *VEGFA* mRNAs was not affected by Lentivirus-N (Fig. 1g). Interestingly, Lentivirus-N-mediated inductions of inflammatory factors were repressed by MCC950 (a specific inhibitor of the NLRP3) (Fig. 1g), indicating that NLRP3 is involved in the induction of proinflammatory cytokines mediated by the N protein.

Moreover, the role of N protein in the regulation of proinflammatory cytokines and chemokines was evaluated in mice. C57BL/6 mice were tail vein-injected with adenovirus-associated virus (AAV)-Lung-enhanced green fluorescent protein (EGFP) (control), AAV-Lung-N, AAV-Lung-N plus MCC950, and AAV-Lung-N plus Ac-YVAD-cmk (a selective irreversible inhibitor of Caspase-1 that blocks IL-1β maturation), respectively. We noted that N protein was expressed in the lung of mice injected with AAV-Lung-N but was not detected in the liver or spleen of mice injected with AAV-Lung-N (Fig. 1h). qRT-PCR analyses showed that *IL-1β*, *IL-6*, *TNF*, *CXLC10*, *CCL2*, *IL-11*, *IL-7*, and *CXCL8* mRNAs were notably induced in the lung of mice injected with AAV-Lung-N, but relatively unaffected in the lung of mice injected with AAV-Lung-CT (Fig. 1i). Interestingly, N-induced expression of *IL-1β*, *IL-6*, *TNF*, *CXLC10*, *CCL2*, *IL-11*, *IL-7*, and *CXCL8* were significantly suppressed by MCC950 and Ac-YVAD-cmk (Fig. 1i).

The role of N protein in the regulation of NF-κB signaling pathway was then determined. Notably, the levels of *NLRP3* mRNA (Supplementary Fig. 1c, top) as well as NLRP3, pro-IL-1β, and pro-IL-18 proteins (Supplementary Fig. 1c, bottom) were relatively unaffected by N protein. These results implicated that N protein regulates NLRP3, pro-IL-1β, and pro-IL-18 independent on NF-κB signaling pathway. We noticed that IL-1β protein promoted the production of inflammatory factors and cytokines, including *IL-6*, *TNF*, *CXCL10*, *IL-11*, *IL-13*, and *CCL2* (Supplementary Fig. 1d). Thus, we speculate that N protein promotes NLRP3 inflammasome activation, thereby inducing mature IL-1β secretion, which subsequently plays a key role in the induction of inflammatory factors and cytokines. Taken together, these results demonstrated that SARS-CoV-2 N protein plays a critical role in the induction of proinflammatory cytokines and suggested that N protein may involve in regulating the activation of the NLRP3 inflammasome.

**IL-1β maturation and IL-6 production are induced by SARS-CoV-2 N protein**. Given the dominant effect of the activator on the expression of the proinflammatory factors, we further determined the role of SARS-CoV-2 N protein in IL-1β maturation. HEK293T-NLRP3 inflammasome system cells were transfected with plasmid encoding SARS-CoV-2 N protein at different concentrations. The results showed that IL-1β secretion (Fig. 2a, top) and IL-1β (p17) cleavage, as well as Caspase-1 (p20) maturation (Fig. 2a, bottom) were induced by N protein in dose-dependent manners. Interestingly, the level of IL-1β protein was significantly induced by the SARS-CoV-2 N protein and slightly enhanced by the SARS-CoV-2 3a protein (Supplementary Fig. 2), confirming a specific role of N protein in the activation of IL-1β protein.

Next, PMA-differentiated THP-1 macrophages were transfected with plasmids encoding SARS-CoV-2 N at different concentrations and treated with Nigericin. We noted that IL-1β secretion was induced by Nigericin and the Nigericin-induced IL-1β secretion was further promoted by N protein in a dose-dependent manner (Supplementary Fig. 3a). Similarly, IL-1β (p17) cleavage and Caspase-1 (p20) maturation were induced by Nigericin, and Nigericin-induced IL-1β (p17) cleavage and Caspase-1 (p20) maturation were further facilitated by N protein in concentration-dependent manners (Supplementary Fig. 3b). However, we found that the level of mature IL-18 was relatively

unaffected by N protein (Supplementary Fig. 3c), although its maturation and production relied on the same cleavage mechanism as IL-1β.

In addition, THP-1 cells stably infected with CT-Lentivirus and N-Lentivirus were differentiated into macrophages, which were then treated with lipopolysaccharide (LPS) plus ATP or LPS plus Nigericin. The results indicated that IL-1β secretion was induced by LPS plus ATP or LPS plus Nigericin, and such inductions were further enhanced significantly by N protein (Fig. 2b, top). Consistently, IL-1β (p17) cleavage was also induced by LPS plus ATP or LPS plus Nigericin, and such inductions were further enhanced by N protein (Fig. 2b, bottom). We noted that the level of mature IL-18 was induced by LPS plus ATP or LPS plus Nigericin, but such inductions were relatively unaffected by N protein (Fig. 2c). Notably, the production of widely described proinflammatory cytokine IL-6 was induced by LPS plus ATP or LPS plus Nigericin, and such induction was also further enhanced by N protein (Fig. 2d). Moreover, IL-6 production was confirmed to be induced by N protein in a dose-dependent manner (Supplementary Fig. 3d).

Next, granulocyte macrophage-colony stimulating factor (GM-CSF) differentiated mice bone marrow-derived monocytes (BMDMs) were infected with CT-Lentivirus or N-Lentivirus and stimulated with LPS, LPS plus ATP, or LPS plus Nigericin. In the treated mouse BMDMs, IL-1β secretion was activated by LPS plus ATP or LPS plus Nigericin, and such activations were further facilitated notably by N protein (Fig. 2e). Similarly, IL-6 production was also induced by LPS plus ATP or LPS plus Nigericin, and such inductions were further promoted by N protein in treated mice BMDMs (Fig. 2f). Therefore, our results demonstrated that SARS-CoV-2 N protein induces IL-1β maturation and IL-6 production.

Therefore, we further investigated the role of NLRP3 involved in IL-1β maturation and IL-6 production induced by the N protein. THP-1 cells stably infected with CT-Lentivirus and N-Lentivirus were differentiated into macrophages, which were pretreated with MCC950 (a specific inhibitor of the NLRP3) and then stimulated with LPS plus ATP or LPS plus Nigericin. Notably, in the presence of LPS plus ATP or LPS plus Nigericin, secreted IL-1β protein was significantly induced by N protein, but N-induced secreted IL-1β protein was remarkably suppressed by MCC950 (Fig. 2g). Similarly, IL-6 protein was enhanced by N protein, whereas N-enhanced IL-6 protein was also repressed by MCC950 (Fig. 2h). These results suggest that N protein induces IL-1β secretion and IL-6 production depending on NLRP3. Moreover, GM-CSF differentiated BMDMs of wide-type mice and *NLRP3*$^{-/-}$ mice were infected with N-Lentivirus and then stimulated with LPS, LPS plus ATP, and LPS plus Nigericin. We noticed that the levels of secreted IL-1β protein (Fig. 2i) and IL-6 protein (Fig. 2j) induced by N protein were significantly lower in the BMDMs of *NLRP3*$^{-/-}$ mice compared with that in BMDMs of wide-type mice (Fig. 2i, j). These data indicate that N protein fails to induce IL-1β protein and IL-6 protein in *NLRP3*$^{-/-}$ mice BMDMs, and thereby suggesting that NLRP3 is required for IL-1β secretion and IL-6 production induced by the N protein. Taken together, our results demonstrate that N protein induces IL-1β maturation and IL-6 production, and reveal that NLRP3 is required for N-induced IL-1β secretion and IL-6 production protein, thereby suggesting that N protein may play an important role in the activation of the NLRP3 inflammasome.

**SARS-CoV-2 N protein interacts with NLRP3 protein**. The role of SARS-CoV-2 N protein in the regulation of NLRP3 inflammasome was then investigated. Initially, the interaction of N protein with the NLRP3 inflammasome components, NLRP3,

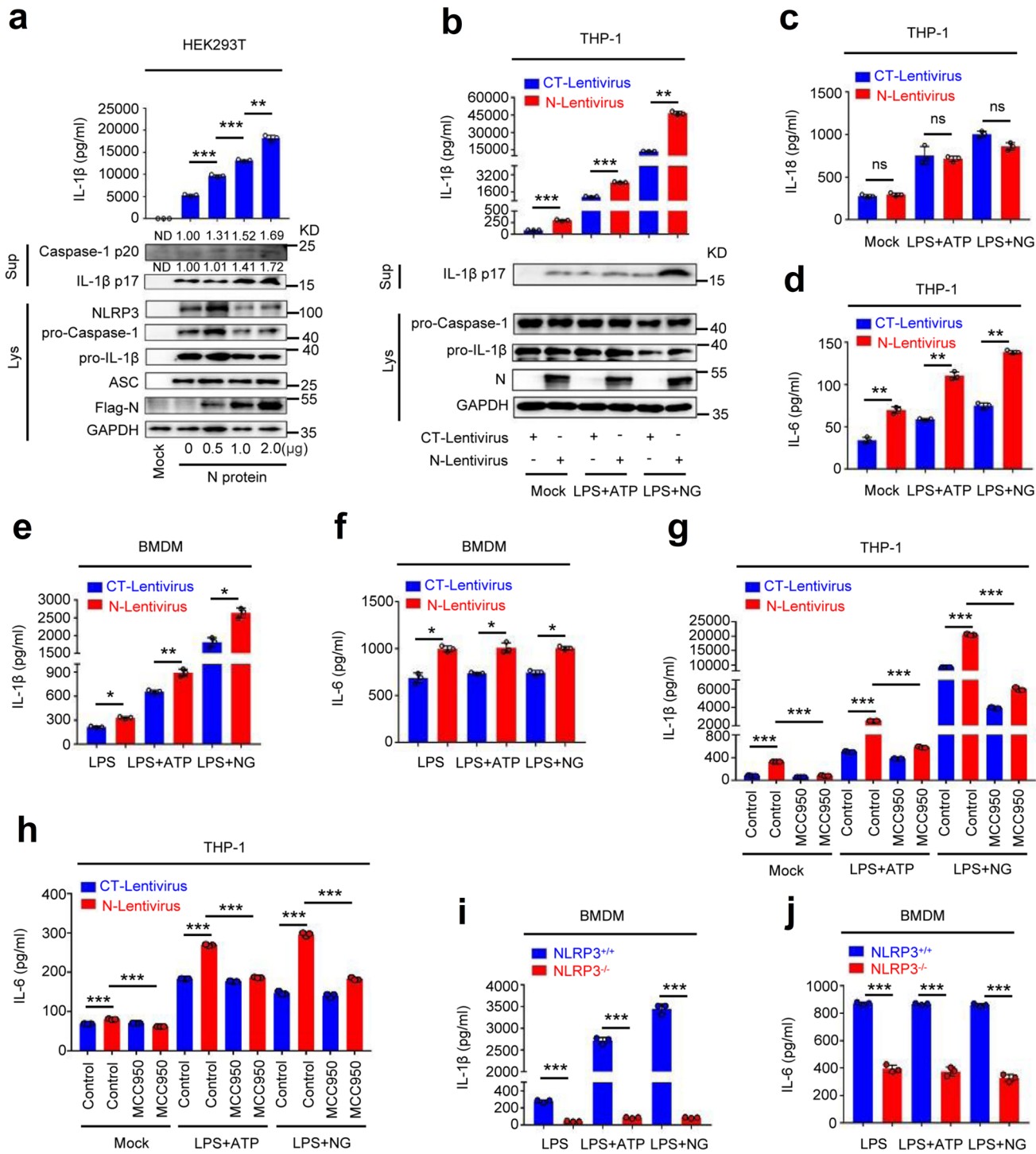

**Fig. 2 IL-1β maturation and IL-6 production are induced by N protein. a** HEK293T cells were co-transfected with plasmids encoding NLRP3, ASC, pro-Caspase-1, and pro-IL-1β, and transfected with plasmids encoding different concentrations of SARS-CoV-2 N for 48 h. Supernatants were analyzed (top) by ELISA for IL-1β. Cell lysates were analyzed (bottom) by immunoblotting. **b–d** THP-1 cells were stably infected with Lentivirus-CT or Lentivirus-N, differentiated into macrophages. Then stimulated with LPS (1 μg/ml) plus ATP (2.5 mM) or LPS (1 μg/ml) plus Nigericin (2 μM). IL-1β (**b**, top), IL-18 (**c**), and IL-6 (**d**) in supernatants was measured by ELISA. Cell lysates were analyzed (**b**, bottom) by immunoblotting. **e, f** GM-CSF-differentiated mice BMDMs were infected with Lentivirus-CT or Lentivirus-N and stimulated with LPS (1 μg/ml), LPS (1 μg/ml) plus ATP (2.5 mM), or LPS (1 μg/ml) plus Nigericin (2 μM). IL-1β (**e**) or IL-6 (**f**) in supernatants was measured by ELISA. **g, h** THP-1 cells were stably infected with Lentivirus-CT or Lentivirus-N, differentiated into macrophages. The cells were treated with MCC950 (0.01 μM) for 1 h and then stimulated with LPS (1 μg/ml) plus ATP (2.5 mM) or LPS (1 μg/ml) plus Nigericin (2 μM). IL-1β protein (**g**) and IL-6 protein (**h**) in supernatants were measured by ELISA. **i, j** GM-CSF-differentiated BMDMs isolated from *NLRP3*[+/+] mice and *NLRP3*[−/−] mice were infected with Lentivirus-N and stimulated with LPS (1 μg/ml), LPS (1 μg/ml) plus ATP (2.5 mM), or LPS (1 μg/ml) plus Nigericin (2 μM). IL-1β protein (**i**) and IL-6 protein (**j**) in supernatants were measured by ELISA. Mock means untreated cells (**a–d** and **g, h**). Data are representative of three independent experiments and one representative is shown. Error bars indicate SD of technical triplicates. Values are mean ± SEM. *$P \leq 0.05$, **$P \leq 0.01$, ***$P \leq 0.001$, two-tailed Student's *t*-test. Source data are provided as a Source Data file.

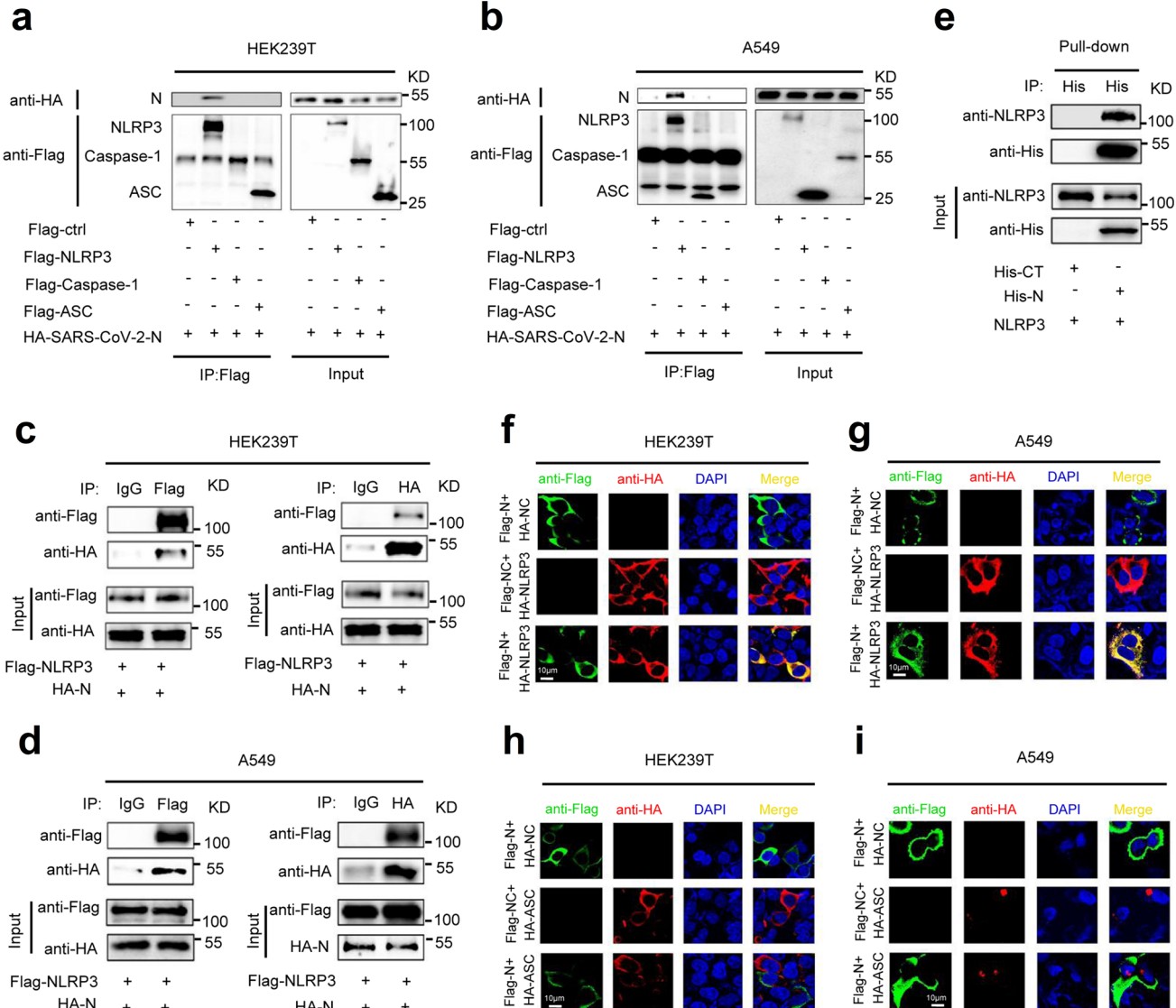

**Fig. 3 SARS-CoV-2 N protein interacts with NLRP3 protein.** HEK293T cells (**a**) or A549 cells (**b**) were co-transfected with HA-SARS-CoV-2-N and Flag-ctrl, Flag-NLRP3, Flag-pro-Caspase-1, or Flag-ASC. Cell lysates were immunoprecipitated using anti-Flag antibody and analyzed using anti-Flag and anti-HA antibody. Cell lysates (40 μg) was used as Input. HEK293T cells (**c**) or A549 cells (**d**) were co-transfected with Flag-NLRP3 and HA-SARS-CoV-2-N. Cell lysates were immunoprecipitated using anti-Flag antibody or anti-HA antibody, and analyzed using anti-Flag and anti-HA antibody. Cell lysates (40 μg) was used as Input. **e** Purified His-CT (10 μg) or His-N (10 μg) was incubated with cell lysates of Flag-NLRP3-transfected HEK293T cells. Cell extracts were incubated with Ni-NTA Agarose beads. Mixtures were analyzed by immunoblotting using anti-NLRP3 or anti-His antibody. Untreated protein includes His-SARS-CoV-2-N (1 μg), His-CT (1 μg), or HEK293T cell lysates were analyzed by immunoblotting (as input). HEK293T cells (**f**, **h**) or A549 cells (**g**, **i**) were transfected with Flag-SARS-CoV-2-N and HA-NLRP3 or HA-ASC for 24 h. Nucleus marker DAPI (blue), Flag-SARS-CoV-2-N (green), and HA-NLRP3 or HA-ASC (red) were then visualized with confocal microscopy. Flag-Ctrl, Flag-NC, or HA-NC means pcDNA3.1(+)−3× flag or pCAGGS-HA empty plasmid. Data were representative of three independent experiments. Source data are provided as a Source Data file.

ASC, and Caspase-1, was explored. Co-immunoprecipitation (Co-IP) assays showed that N protein only interacted with NLRP3 protein and failed to interact with ASC protein or Caspase-1 protein in HEK293T and A549 cells (Fig. 3a, b). Reciprocal Co-IP assays further conformed that NLRP3 protein interacted with N protein (Fig. 3c, d), but failed to interact with 3a protein (Supplementary Fig. 4a, b) in HEK293T and A549 cells. Importantly, we demonstrated that purified His-N protein could directly bind to NLRP3 protein (Fig. 3e). There are four types of ASC-dependent inflammasomes (NLRP1, NLRP3, NLRC4, and AIM2) that have been reported[12]. Here we determined the role of N protein in the regulation of NLRP1, NLRP3, NLRC4, and AIM2 inflammasomes. Co-IP assays showed that N

protein only interacted with the NLRP3 protein, but failed to interact with the NLRP1, NLRC4, or AIM2 proteins (Supplementary Fig. 5a). Notably, we showed that N protein could induce IL-1β production in the presence of NLRP3, but failed to promote IL-1β production in the presence of NLRP1, NLRC4, or AIM2 (Supplementary Fig. 5b). These results demonstrate that N protein specifically and directly triggers NLRP3. Moreover, confocal microscope analyses showed that in HEK293T and A549 cells, N protein and NLRP3 protein were co-localized in the cytoplasm (Fig. 3f, g and Supplementary Movie 1), and N protein failed to interact with ASC (Fig. 3h, i and Supplementary Movie 2, 3). Taken together, these results demonstrated that N protein

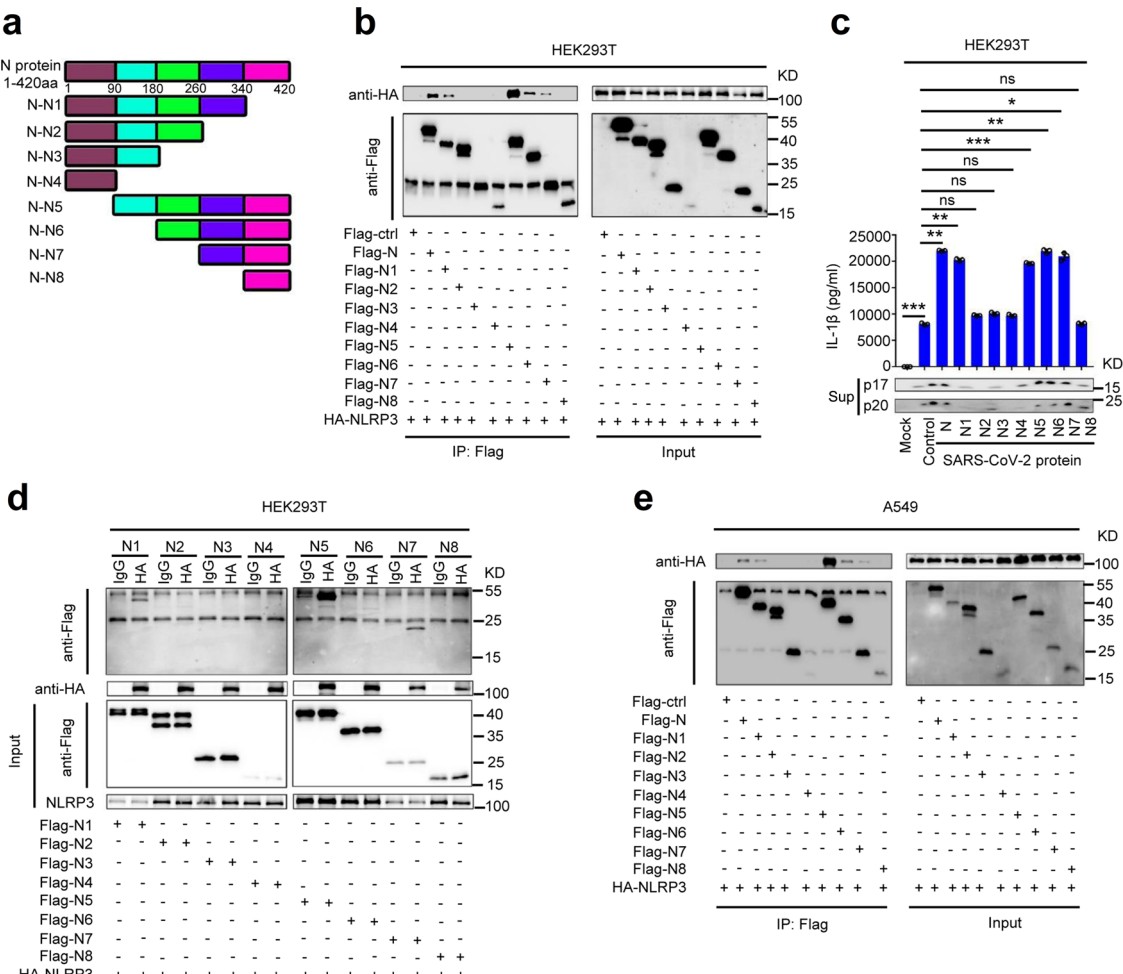

**Fig. 4 Sequence of N protein involved in NLRP3 inflammasome activation. a** Schematic diagram of wild-type SARS-CoV-2-N protein and truncated mutants N protein (N1 to N8). HEK293T cells (**b**) or A549 cells (**d**) were co-transfected with HA-NLRP3 and Flag-Ctrl, Flag-N truncated mutants (N1–N8). Cell lysates were immunoprecipitated using anti-Flag antibody and analyzed using anti-Flag and anti-HA antibody. Cell lysates (40 μg) were used as Input. **c** HEK293T cells were co-transfected with HA-NLRP3 and Flag-N truncated mutants (N1–N8). Cell lysates were immunoprecipitated using anti-HA antibody, IgG antibody was used as negative control, and analyzed using anti-Flag and anti-HA antibody. Cell lysates (40 μg) was used as Input. **e** HEK293T cells were co-transfected with plasmids encoding NLRP3, ASC, pro-Casp1, and pro-IL-1β, and transfected with plasmids encoding SARS-CoV-2-N protein and truncated mutants N protein (N1–N8) for 48 h. Supernatants were analyzed by ELISA for IL-1β and by WB for p17 and p20. Flag-ctrl means pcDNA3.1(+)–3× flag empty plasmid. Mock means untreated cells (**e**). Control means transfected empty plasmid (**e**). Data are representative of three independent experiments and one representative is shown. Error bars indicate SD of technical triplicates. Values are mean ± SEM. *$P \leq 0.05$, **$P \leq 0.01$, ***$P \leq 0.001$, two-tailed Student's $t$-test. Source data are provided as a Source Data file.

specifically interacts with NLRP3 protein to regulate the NLRP3 inflammasome.

**Sequence of N protein involved in NLRP3 inflammasome activation.** The sequences of SARS-CoV-2 N protein involved in the interaction with NLRP3 protein were identified by evaluating progressive truncated mutants of N protein, N1–N8 (Fig. 4a). Co-IP results showed that NLRP3 interacted with N1 (1-340), N5 (90-420), N6 (180-420), and N7 (260-420), but failed to interact with N2 (1-260), N3 (1-180), N4 (1-90), and N8 (340-420) in HEK293T cells (Fig. 4b, c) and A549 cells (Fig. 4d), suggesting that the domain containing 260aa–340aa of N protein is involved in the interaction with NLRP3.

Next, the effect of N protein progressive truncations (N1–N8) on the activation of the NLRP3 inflammasome was evaluated. The HEK293T-NLRP3 inflammasome system cells were transfected with plasmids encoding N protein or its truncated mutants N1–N8, respectively. ELISA assays showed that IL-1β secretion, IL-1β p17 cleavage, and Caspase-1 p20 maturation in the cell

supernatants were significantly induced by the N1, N5, N6, and N7, but not affected by N2, N3, N4, and N8 (Fig. 4e), indicating that the domain containing 260aa–340aa of N protein is involved in the activation of the NLRP3 inflammasome. Taken together, we demonstrated that the sequence 260aa–340aa of SARS-CoV-2 N interacts with NLRP3 to activate the NLRP3 inflammasome.

**Interaction of NLRP3 with ASC is promoted by N protein.** As SARS-CoV-2 N interacts with NLRP3, we would like to know the role of N in regulating the components of the NLRP3 inflammasome. Initially, the effect of N protein on the expression of NLRP3, Caspase-1, and ASC was examined. The results showed that the protein levels of NLRP3, Caspase-1, and ASC were not influenced by N protein in HEK293T cells (Fig. 5a). However, N and ASC could immunoprecipitate with each other in the presence of NLRP3 in HEK293T cells (Fig. 5b, c) and A549 cells (Fig. 5d, e), suggesting that N, NLRP3, and ASC three proteins together might form a complex N-NLRP3-ASC. Interestingly, the interaction of NLRP3 and ASC was enhanced by N protein in

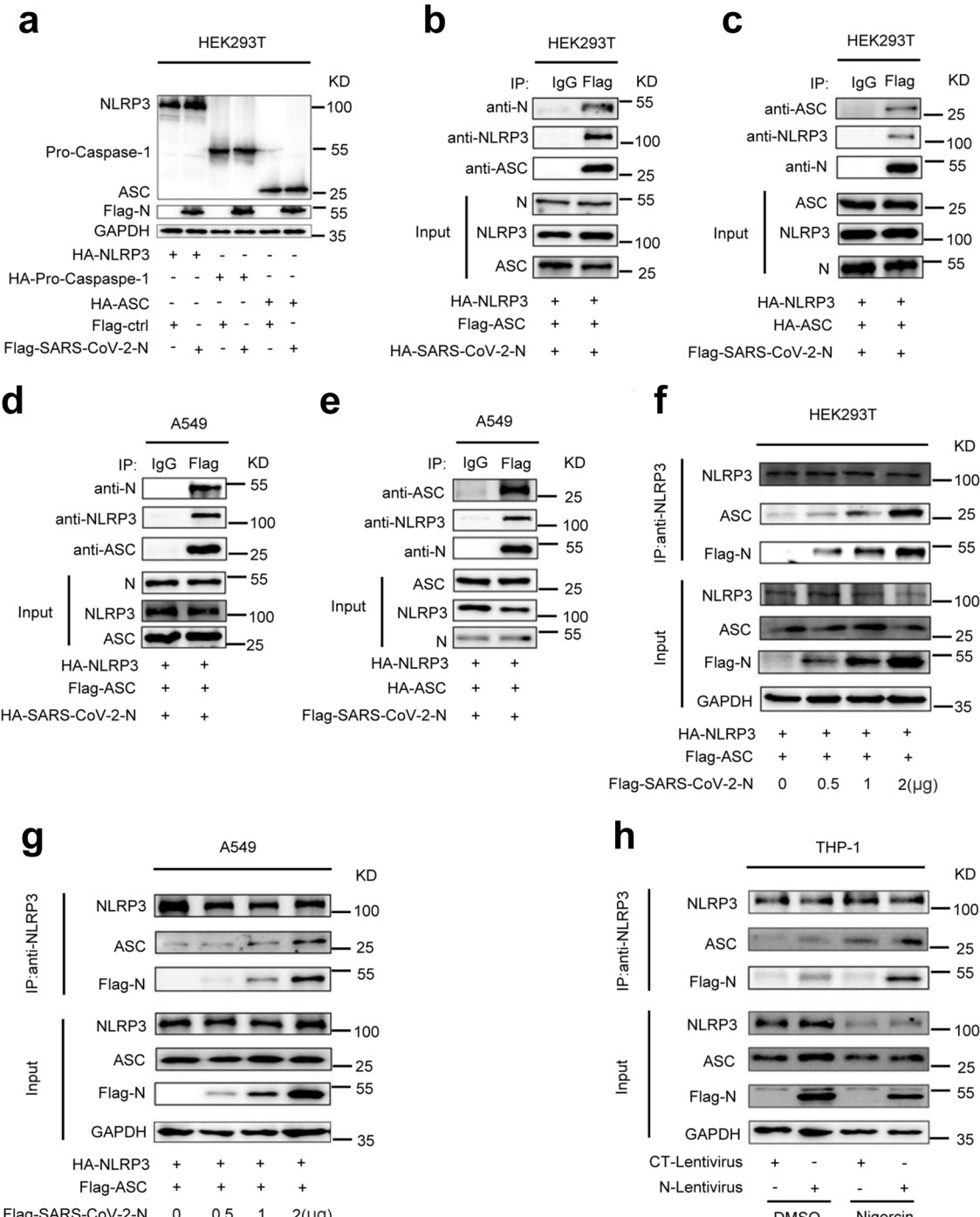

**Fig. 5 Interaction of NLRP3 with ASC is promoted by N protein. a** HEK293T cells were co-transfected with Flag-Ctrl or Flag-SARS-CoV-2-N plus HA-NLRP3, HA-pro-Casp1, or HA-ASC for 36 h, the indicated proteins in cell extract were analyzed by WB. HEK293T cells (**b**) or A549 cells (**d**) were co-transfected with HA-SARS-CoV-2-N, HA-NLRP3, and Flag-ASC for 24 h. Cell lysates were immunoprecipitated using anti-Flag antibody and analyzed using anti-NLRP3, anti-ASC, and anti-N antibody. HEK293T cells (**c**) or A549 cells (**e**) were co-transfected with Flag-SARS-CoV-2-N, HA-NLRP3, and HA-ASC for 24 h. Cell lysates were immunoprecipitated using anti-Flag antibody and analyzed using anti-NLRP3, anti-ASC, and anti-N antibody. Cell lysates (40 μg) was used as Inputs. HEK293T cells (**f**) or A549 cells (**g**) were co-transfected with different concentration of Flag-SARS-CoV-2-N plus HA-NLRP3 and Flag-ASC for 24 h. Cell lysates were immunoprecipitated using anti-NLRP3 antibody and analyzed using anti-NLRP3, anti-ASC, anti-GAPDH, and anti-Flag antibody. **h** THP-1 macrophages were stably infected with Lentivirus-CT or Lentivirus-N, and stimulated with 2 μM Nigericin or DMSO for 2 h. Cell lysates were immunoprecipitated using anti-NLRP3 antibody and analyzed using anti-NLRP3, anti-ASC, anti-GAPDH, and anti-N antibody. Flag-ctrl means pcDNA3.1(+)–3× flag empty plasmid. Data are representative of three independent experiments. Source data are provided as a Source Data file.

dose-dependent manners in HEK293T cells (Fig. 5f) and A549 cells (Fig. 5g), but 3a protein had no influence in the interaction of NLRP3 and ASC compared with N protein (Supplementary Fig. 6a, b). Co-IP results also demonstrated that N protein interacted with endogenous ASC in the presence of NLRP3 and the interaction was promoted by stimulating with Nigericin in PMA-differentiated THP-1-N cells (Fig. 5h). Taken together, these results suggested that SARS-CoV-2 N promotes the interaction of NLRP3 with ASC.

**SARS-CoV-2 N protein facilitates NLRP3 inflammasome assembly**. The role of SARS-CoV-2 N protein in regulating the assembly of the NLRP3 inflammasome was then explored. Localization of NLRP3 as a speck structure in the cytoplasm is an indicator of inflammasome complex formation[27]. In HEK293T cells (Fig. 6a, top) and A549 cells (Fig. 6a, bottom), NLRP3 alone was diffusely distributed in the cytoplasm, whereas in the presence of SARS-CoV-2 N protein, NLRP3 co-localized with N and formed specks. However, in the presence of SARS-CoV-2 3a protein, NLRP3 was co-localized with 3a, but failed to form specks (Fig. 6a). We noted that N2 (1aa–260aa) (Fig. 6b) and N8 (340aa–420aa) (Fig. 6c) were distributed in both the nucleus and cytoplasm, and failed to co-localize with NLRP3 and NLRP3 failed to form specks in the cytoplasm of HEK293T cells (Fig. 6b, c, top) and A549 cells (Fig. 6b, c, bottom). Confocal microscope showed that in PMA-differentiated THP-1 macrophages, N protein alone was diffusely distributed in the cytoplasm (Fig. 6d); endogenous NLRP3 protein alone also diffusely located in the cytoplasm (Fig. 6e, top). However, in the presence of N protein, endogenous NLRP3 protein aggregated and formed specks in the cytoplasm (Fig. 6e, bottom). Collectively, these data suggested that SARS-CoV-2 N protein may promote the formation of the NLRP3 inflammasome complex.

In the process of inflammasome assembly, ASC protein is aggregated to form oligomers, which is required for Caspase-1 activation[28]. Notably, SARS-CoV-2 N protein specifically promoted ASC oligomerization mediated by NLRP3, but had no effect on ASC oligomerization induced by NLRC4 or AIM2 (Supplementary Fig. 7a). We also showed that SARS-CoV-2 N protein significantly induced ASC oligomerization, but SARS-CoV-2 3a protein had no effect on ASC oligomerization (Supplementary Fig. 7b). Interestingly, in PMA-differentiated THP-1 macrophages (Fig. 6f) and GM-CSF differentiated BMDMs (Fig. 6g), oligomerization of endogenous ASC protein was stimulated by Nigericin and Nigericin-induced ASC oligomerization was further enhanced in the presence of N protein (Fig. 6f, g). Interestingly, we noticed that N protein-induced endogenous ASC oligomerization was significantly inhibited by MCC950 in PMA-differentiated THP-1 macrophages (Supplementary Fig. 7c) and GM-CSF-differentiated NLRP3$^{-/-}$ BMDMs (Supplementary Fig. 7d). In addition, HEK293T-NLRP3 inflammasome system cells were transfected with plasmids encoding SARS-CoV-2-N protein and its truncated mutants, respectively. Notably, ASC oligomerization was activated by N1, N5, N6, and N7 proteins, but not influenced by N2, N3, N4, and N8 proteins (Fig. 6h), demonstrating that the sequence 260aa–340aa of N protein interacts with NLRP3 to promote ASC oligomerization.

Moreover, we explored the effect of N protein on the assembly of the NLRP3 inflammasome complex. HEK293T cells (Fig. 7a) and A549 cells (Fig. 7b) were co-transfected with plasmids as indicated. In the presence of GFP and NLRP3, GFP was distributed in both the cytoplasm and nucleus, NLRP3 was diffusely distributed in the cytoplasm (Fig. 7a (a–e) and Fig. 7b (a–e)). In the presence of N and NLRP3, NLRP3 was co-localized

with N and formed speck structures (Fig. 7a (f–j) and Fig. 7b (f–j)). In the presence of GFP and ASC, GFP was distributed in both the cytoplasm and nucleus, ASC was distributed in the nucleus and cytoplasm, and formed small ring structures (Fig. 7a (k–o) and Fig. 7b (k–o)). When N and ASC presented together, N and ASC were not co-localized (Fig. 7a (p–t) and Fig. 7b (p–t)). Notably, in the presence of GFP, ASC, and NLRP3, NLRP3 and ASC were co-localized in the cytoplasm to form "ring-like" structures (Fig. 7a (u–y) and Fig. 7b (u–y)). Importantly, when N, ASC, and NLRP3 expressed together, the three proteins were obviously co-localized to form "sphere-like" structures (Fig. 7a (z–ad) and Fig. 7b (z–ad), and Supplementary Movies 4 and 5). However, in the presence of 3a, ASC, and NLRP3 together, the three proteins were co-localized, but failed to form "sphere-like" structures (Fig. 7a (ae–ai) and Fig. 7b (ae–ai)).

Previous studies demonstrated that the NIMA-related kinase 7 (NEK7) is important to license NLRP3 inflammasome activation through directly binding to NLRP3[29]. Here we investigated that the role of NEK7 in NLRP3 inflammasome activation is mediated by SARS-CoV-2 N protein. Small interfering RNAs specifically targeting NEK7 were generated. The results showed that *NEK7* mRNA (Supplementary Fig. 8a–c, top) and NEK7 protein (Supplementary Fig. 8a–c, bottom) were significantly attenuated by si-*NEK7*−3 but relatively unaffected by si-*NEK7*−1 and si-*NEK7*−2 in THP-1 cells (Supplementary Fig. 8a–c). THP-1 cells stably infected N-Lentivirus were differentiated into macrophages and then transfected with si-*NEK7*−3 and treated with Nigericin. Notably, IL-1β protein was induced by N protein and significantly stimulated by Nigericin, whereas N-promoted and Nigericin-induced IL-1β proteins were relatively not affected by si-*NEK7*−3 (Supplementary Fig. 8d). Similarly, N-promoted oligomerization of ASC in THP-1 cells (Supplementary Fig. 8e) and N-facilitated interaction of NLRP3-ASC were not influenced by si-*NEK7*−3 in THP-1 cells (Supplementary Fig. 8f), HEK293T cells (Supplementary Fig. 8g), and A549 cells (Supplementary Fig. 8h). These results indicated that NEK7 is not required for N protein-mediated NLRP3 inflammasome activation and NLRP3-ASC interaction. Taken together, these data demonstrated that SARS-CoV-2 N protein, but not SARS-CoV-2 3a protein, promotes the assemble of the NLRP3 inflammasome complex.

**N protein induces lung injury in mice by activating NLRP3 inflammasome**. The biological role of SARS-CoV-2 N protein in activating the NLRP3 inflammasome was then assessed. As the main clinical symptom of COVID-19 is acute lung injury[5], the "cytokine storm" is considered to be an important cause of acute lung injury[26]. Therefore, we explored the effect of N protein on lung injury using an AAV-lung-N C57BL/6 mouse model. C57BL/6 mice were subjected to tail vein injection with AAV-Lung-EGFP or AAV-Lung-N. The results indicated that the protein levels of IL-1β and IL-6 in the sera were significantly higher in AAV-N-infected mice than that in AAV-CT-infected mice (Supplementary Fig. 9a, b). In addition, C57BL/6 mice injected with AAV-Lung-EGFP or AAV-Lung-N were subjected to intraperitoneal injection with or without Ac-YVAD-cmk and MCC950. We noted that the protein levels of IL-1β and IL-6 were much higher in the sera of AAV-N-infected mice as compared to AAV-CT-infected mice. However, AAV-N-mediated inductions of IL-1β and IL-6 proteins were repressed by Ac-YVAD-cmk (Supplementary Fig. 9c, d) and MCC950 (Fig. 8a, b). We noticed that unlike IL-1β and IL-6, the IL-18 protein was not affected by AAV-N in the sera pretreated with MCC950 (Fig. 8c). Notably, immunohistochemical fluorescence analyses showed that IL-1β protein (Fig. 8d) and IL-6 protein (Fig. 8e) were highly expressed in the lungs of mice carrying AAV-Lung-N, whereas such

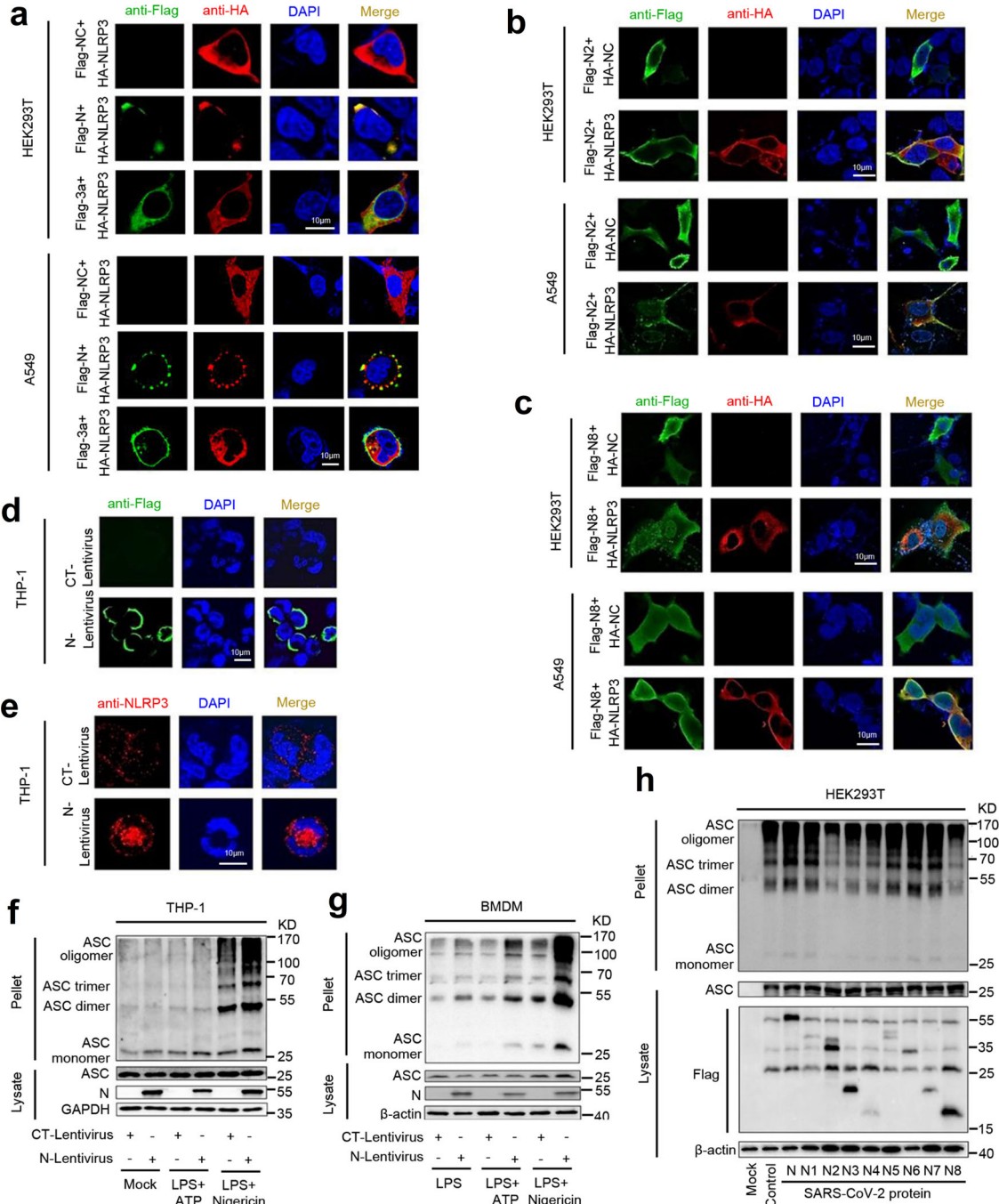

**Fig. 6 N protein facilitates NLRP3 aggregation and ASC oligomerization. a** HEK293T cells (top) or A549 cells (bottom) were transfected with Flag-NC/ HA-NLRP3, Flag-N/HA-NLRP3, or Flag-3a/HA-NLRP3 for 24 h. The sub-cellular locations of NLRP3 (red), Nucleus marker DAPI (blue), and SARS-CoV-2 N or SARS-CoV-2 3a (green) were visualized with confocal microscopy. **b** HEK293T cells (top) or A549 cells (bottom) were transfected with Flag-N2+HA-NC or Flag-N2+HA-NLRP3 for 24 h. The sub-cellular locations of NLRP3 (red), Nucleus marker DAPI (blue), and SARS-CoV-2-N8 (green) were visualized with confocal microscopy. **c** HEK293T cells (top) or A549 cells (bottom) were transfected with Flag-N8+HA-NC or Flag-N8+HA-NLRP3 for 24 h. The sub-cellular locations of NLRP3 (red), Nucleus marker DAPI (blue), and SARS-CoV-2-N8 (green) were visualized with confocal microscopy. **d**, **e** PMA-differentiated THP-1 macrophages were stably infected with Lentivirus-CT or Lentivirus-N. The sub-cellular locations of SARS-CoV-2-N (green, **d**) Nucleus marker DAPI (blue), and NLRP3 (red, **e**) were visualized with confocal microscopy. **f** THP-1 macrophages were stably infected with Lentivirus-CT or Lentivirus-N, and stimulated by LPS (1 μg/ml) plus ATP (2.5 mM) or LPS (1 μg/ml) plus Nigericin (2 μM). ASC oligomerization was analyzed by immunoblotting. **g** GM-CSF differentiated mice BMDMs were infected with Lentivirus-CT or Lentivirus-N and stimulated with LPS (1 μg/ml), LPS (1 μg/ml) plus ATP (2.5 mM), or LPS (1 μg/ml) plus Nigericin (2 μM). ASC oligomerization was analyzed by immunoblotting. **h** HEK293T cells were co-transfected with plasmids encoding NLRP3, ASC, pro-Caspase-1, or pro-IL-1β, and transfected with plasmids encoding SARS-CoV-2-N protein and truncated mutants N protein (N1 to N8) for 48 h, ASC oligomerization was analyzed by immunoblotting. Flag-NC or HA-NC means pcDNA3.1(+)–3× flag or pCAGGS-HA empty plasmid (**a–c**). Mock means untreated cells (**f**, **h**). Control means transfected empty plasmid (**h**). Data are representative of three independent experiments. Source Data are provided as a Source Data file.

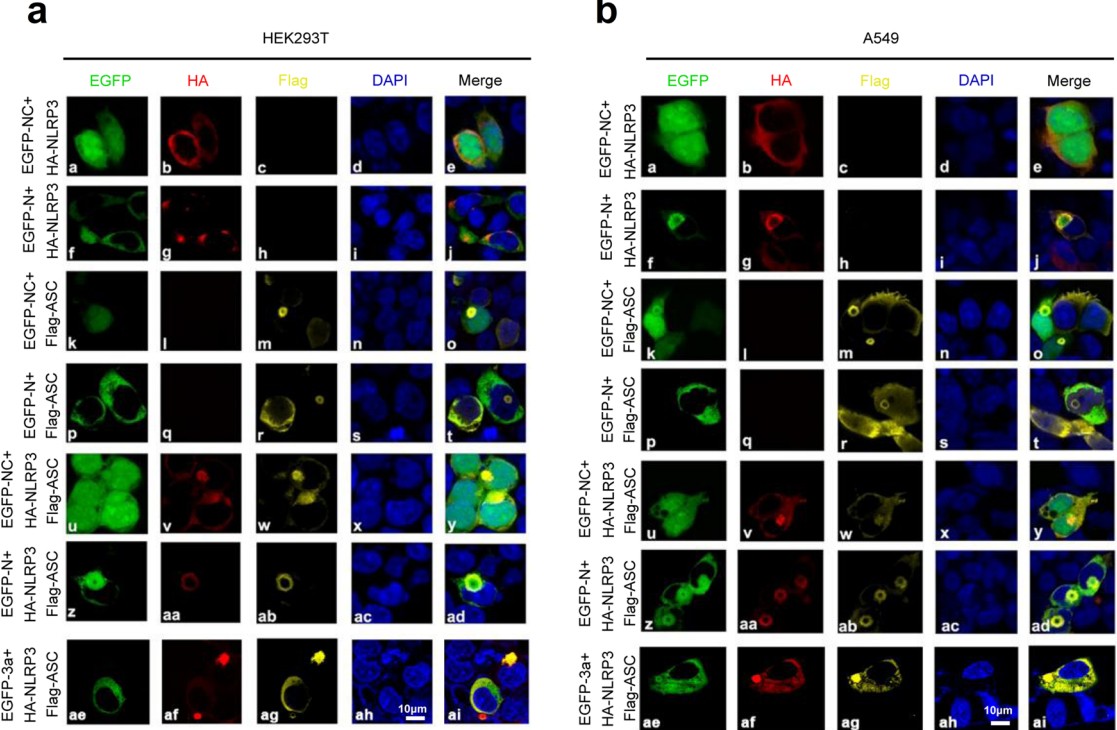

**Fig. 7 N protein promotes NLRP3 inflammasome assembly.** HEK293T cells (**a**) and A549 cells (**b**) were co-transfected with GFP/HA-NLRP3, GFP-N/HA-NLRP3, GFP/Flag-ASC, GFP-N/Flag-ASC, GFP/HA-NLRP3/Flag-ASC, GFP-N/HA-NLRP3/Flag-ASC, and GFP-3a/HA-NLRP3/Flag-ASC for 24 h. The subcellular locations of HA-tagged NLRP3 (red), Flag-tagged ASC (yellow), GFP-tagged N or GFP-tagged 3a (green), and nucleus marker DAPI (blue) were visualized with confocal microscopy. GFP-NC means pEGFP-C1 empty plasmid (**a**, **b**). Data are representative of three independent experiments. Source data are provided as a Source Data file.

activations were attenuated by Ac-YVAD-cmk and MCC950 (Fig. 8d, e). Interestingly, hematoxylin and eosin staining analyses indicated that inflammatory lesions and tissue injuries were obvious in the lungs of mice carrying AAV-Lung-N, but such pathological changes were repressed by the administration of Ac-YVAD-cmk and MCC950 (Fig. 8f). These results suggest that the NLRP3 inflammasome is required for the development of mice lung injury induced by the N protein.

The role of NLRP3 in N-mediated mice lung injury was further determined by in C57BL/6 $NLRP3^{-/-}$ mice. Knockout of $NLRP3$ gene in C57BL/6 $NLRP3^{-/-}$ mice was confirmed by genotyping using mouse tail DNA samples (Supplementary Fig. 9e). C57BL/6 $NLRP3^{+/+}$ and C57BL/6 $NLRP3^{-/-}$ mice were tail vein-injected with AAV-Lung-EGFP (control) and AAV-Lung-N, respectively. The results confirmed that N protein was specifically expressed in the lung of mice injected with AAV-Lung-N, but not detected in the lung of mice injected with AAV-Lung-EGFP (Supplementary Fig. 9f). ELISA results showed that IL-1β protein and IL-6 protein were significantly induced by AAV-N in the sera of $NLRP3^{+/+}$ mice, but not affected by AAV-N in the sera of $NLRP3^{-/-}$ mice (Fig. 9a, b), indicating that NLRP3 is essential for N-mediated activation of IL-1β and IL-6 proteins. We noticed that unlike IL-1β and IL-6, IL-18 protein was not affected by AAV-N in both $NLRP3^{+/+}$ and $NLRP3^{-/-}$ mice (Fig. 9c). In addition, immuno-histochemical fluorescence analyses showed that IL-1β and IL-6 proteins were highly expressed in the lungs of $NLRP3^{+/+}$ mice carrying AAV-Lung-N, whereas such activations were decreased in the lungs of $NLRP3^{-/-}$ mice carrying AAV-Lung-N (Fig. 9d, e). Hematoxylin and eosin staining analyses indicated that inflammatory lesions and tissue injuries were obvious in the lungs of $NLRP3^{+/+}$ mice carrying AAV-Lung-N, but such pathological changes were repressed in the lungs of $NLRP3^{-/-}$

mice carrying AAV-Lung-N (Fig. 9f). Collectively, these results suggested that N protein induces a systemic inflammation and lung injury in mice via activating the NLRP3 inflammasome.

Moreover, C57BL/6 mice were tail vein-injected with AAV-Lung-EGFP or AAV-Lung-N, treated with phosphate-buffered saline (PBS) or LPS, and intraperitoneally injected with or without MCC950 or Ac-YVAD-cmk, as indicated. Interestingly, the survival rates were reduced in mice treated with LPS; relatively unchanged between mice treated with LPS plus injected with AAV-Lung-EGFP and mice treated with LPS; further downregulated significantly in mice treated with LPS plus injected with AAV-Lung-N; and recovered in mice treated with LPS plus injected with AAV-Lung-N and treated with MCC950 (Fig. 8g) or Ac-YVAD-cmk (Supplementary Fig. 9g). Taken together, we reveal a pathological mechanism of COVID-19, in which SARS-CoV-2 N protein promotes inflammation and lung injury via promoting the activation of the NLRP3 inflammasome (Fig. 10).

## Discussion

SARS-CoV-2 infection could cause acute lung injury and one of the important reasons is dysregulation of the immune system in the lung[7,8] that results in the release of a large number of inflammatory factors and forms "cytokine storm"[9]. In this study, we found that SARS-CoV-2 infection of macrophages and DCs would promote the expression of abundant cytokines and chemokines by analyzing the GEO database (GSE155106). Next, through screening, we found that SARS-CoV-2 N protein promoted the expression of inflammatory factors by activating NLRP3 inflammasome, resulting in mouse lung injury and exacerbating the death of mice in the sepsis model. Further studies indicated that N protein directly interacted with NLRP3 to

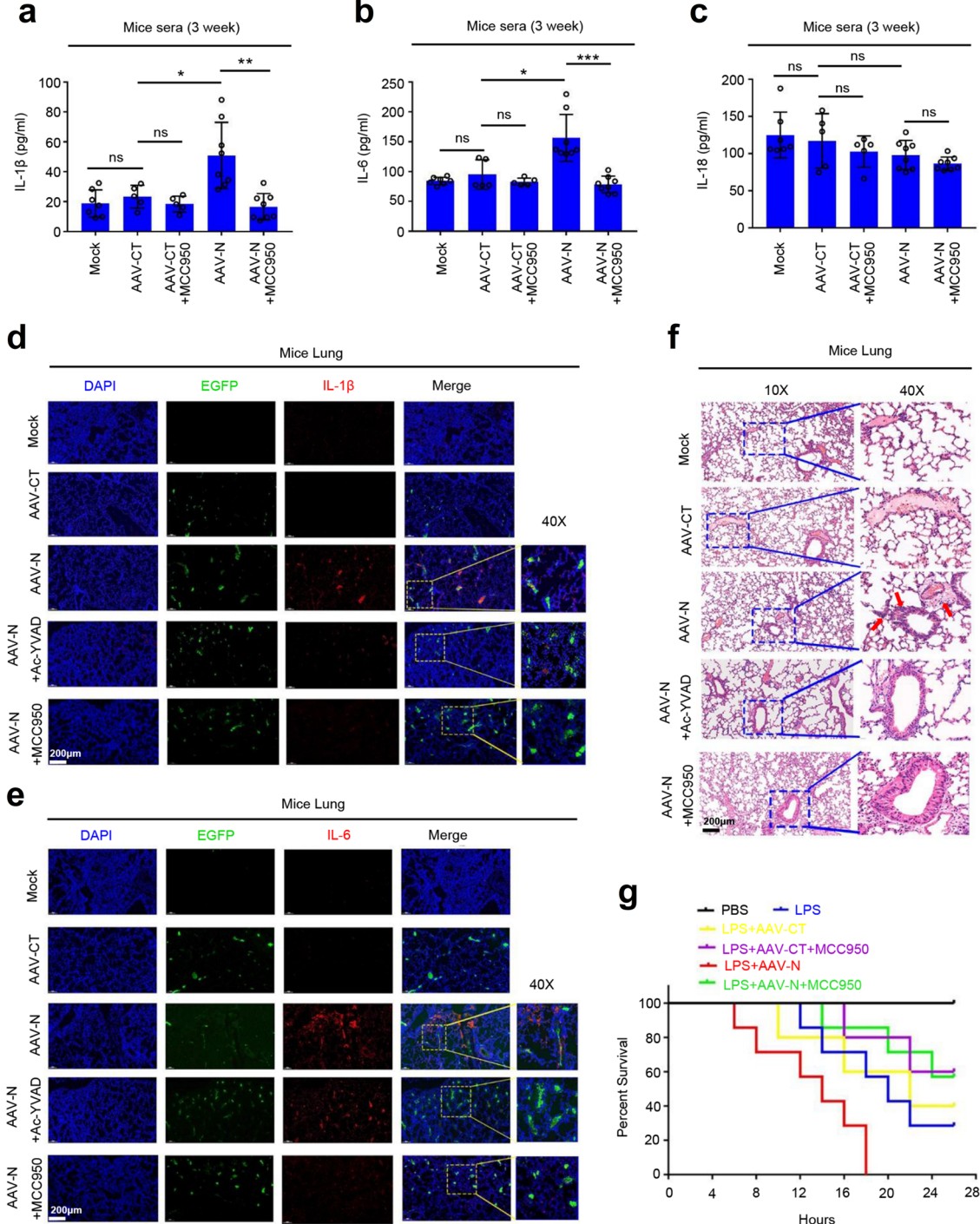

**Fig. 8 N protein induces lung injury in mice by activating NLRP3 inflammasome. a–c** C57BL/6 genetic background mice were tail vein-injected with 300 μl containing $5 \times 10^{11}$ vg of AAV-Lung-EGFP ($n = 10$) or AAV-Lung-N ($n = 16$) for 2 weeks and then treated with MCC950 (50 mg/kg) by intraperitoneal injection for AAV-Lung-EGFP ($n = 5$) or AAV-Lung-N ($n = 8$) mice. Serum was collected at 3 weeks for each group from the orbit. IL-1β (**a**), IL-6 (**b**), or IL-18 (**c**) in the sera was measured by ELISA. Points represent the value of each serum samples. **d–f** At 3 weeks, mice were killed and the lungs were collected. Histoimmunofluorescence analysis of IL-1β (**d**, red) or IL-6 (**e**, red) in the lung after AAV-CT or AAV-N infection. Scale bar is 200 μm (10×) or 50 μm (40×). Histopathology analysis of lung after AAV-CT or AAV-N infection (**f**). Red arrows indicated the infiltrated inflammatory cells. Scale bar is 200 μm (10×) or 50 μm (40×). **g** After 4 weeks, pretreated AAV-Lung-N mice ($n = 7$) with DMSO, pretreated AAV-Lung-N ($n = 7$) mice with MCC950, or pretreated AAV-Lung-EGFP ($n = 5$) with MCC950. After 30 min, mock group ($n = 7$), AAV-Lung-EGFP group ($n = 5$), AAV-Lung-N ($n = 7$) group, and another AAV-Lung-N ($n = 7$) group were treated with LPS (30 mg/kg) by intraperitoneal injection. Another mock group ($n = 3$) was intraperitoneally injected with PBS. The mice survival rates were evaluated every 2 h post treatment. Mock means inject the same dose of PBS as other groups (**a–h**). Data are representative of two independent experiments and one representative is shown. Error bars indicate SD of each serum samples, *$P \leq 0.05$, **$P \leq 0.01$, ***$P \leq 0.001$, two-tailed Student's $t$-test (**a–c**). One-way ANOVA analysis (**g**). Source data are provided as a Source Data file.

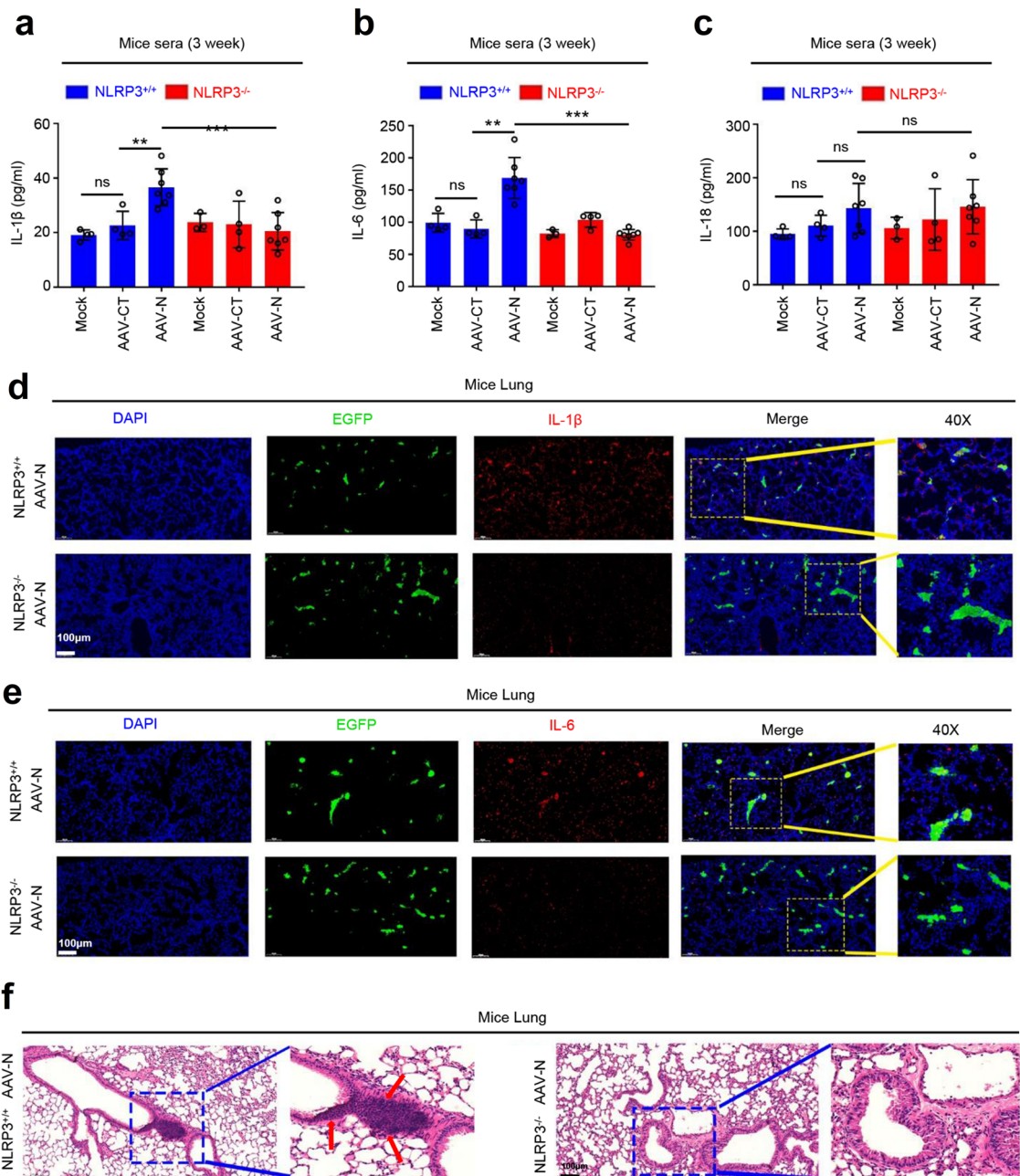

**Fig. 9 NLRP3 has an important function in N protein-induced lung injury. a–c** $NLRP3^{+/+}$ C57BL/6 mice or $NLRP3^{-/-}$ C57BL/6 mice were tail vein-injected with 300 μl containing $5 \times 10^{11}$ vg of AAV-Lung-EGFP ($n = 4$) or AAV-Lung-N ($n = 7$). Serum was collected at 3 weeks for each group from the orbit. IL-1β (**a**), IL-6 (**b**), or IL-18 (**c**) in the sera were measured by ELISA assay. Points represent the value of each serum samples. **d, e** Histoimmunofluorescence analysis of IL-1β (**d**, red) or IL-6 (**e**, red) in the lung after AAV-N infected $NLRP3^{+/+}$ mice or $NLRP3^{-/-}$ mice. Scale bar is 200 μm (10×) or 50 μm (40×). **f** Histopathology analysis of lung after AAV-N infected $NLRP3^{+/+}$ mice or $NLRP3^{-/-}$ mice. Red arrows indicated the infiltrated inflammatory cells. Scale bar is 200 μm (10×) or 50 μm (40×). Data are representative of two independent experiments and one representative is shown. Error bars indicate SD of each serum samples, *$P \leq 0.05$, **$P \leq 0.01$, ***$P \leq 0.001$, two-tailed Student's $t$-test (**a–c**). Source data are provided as a Source Data file.

promote the assembly of NLRP3 inflammasome and thereby activated NLRP3 inflammasome. Above all, our results demonstrated a novel mechanism by which SARS-CoV-2 N protein induced lung injury through NLRP3 inflammasome.

As an important part of the innate immune system, NLRP3 inflammasome plays a crucial role in the recognition of cell damage and pathogenic microbial infection[10–13]. Our previous studies found that M protein of Dengue virus caused tissue damage and vascular leakage in mice by activating NLRP3

inflammasome[14], whereas NS5 protein of Zika virus promoted virus entry into the brain, which aggravated the death of mice by activating NLRP3 inflammasome[30]. Whether NLRP3 inflammasome plays a vital role in COVID-19? In this work, we found that SARS-CoV-2 N protein could promote the expression of inflammatory factors. However, after adding MCC950, a specific inhibitor of the NLRP3 or Ac-YVAD-cmk, an inhibitor of the Caspase-1[31], the levels of N protein-induced inflammatory factors were restored. N protein, as the main structural protein of SARS-

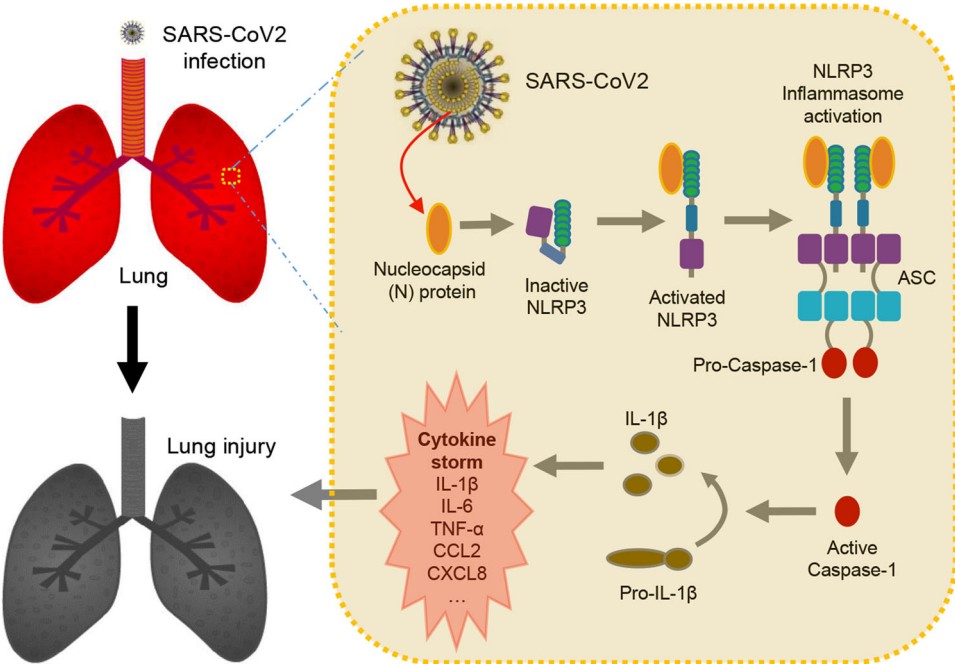

**Fig. 10 Model by which N protein induces lung injury by activating NLRP3 inflammasome.** During SARS-CoV-2 infection, the nucleocapsid protein (N) could directly interact with NLRP3 to promote the assembly and activation of NLRP3 inflammasome, thus leading to the production of a large number of inflammatory factors (such as IL-1β, IL-6, TNF, CCL2, and CCL8) and the occurrence of lung injury in mice.

CoV-2, is involved in virus replication, assembly, and immune regulation[32,33]. As the most stably expressed structural protein, it is the gold standard in nucleic acid test of SARS-CoV-2. Recent research has also found that N protein combats host RNA interference-mediated antiviral response through its double-stranded RNA-binding activity[34] and N protein can induce both humoral and cellular immune response after infection[35]. Our results demonstrate that SARS-CoV-2 N protein, as a part of the viral envelop protein, is exited in the cytoplasm to trigger NLRP3 activation prior to viral assembly and increases the risk of activating the acute inflammatory response.

As important inflammatory factors in the innate immune system, IL-1β and IL-6 play important roles in the host defense against the virus[36]. Our previous studies have shown that Dengue virus promotes vascular leakage in mice by inducing IL-1β production[37]. Moreover, IL-1β is a highly potent proinflammatory mediator that induces vasodilation and attracts granulocytes to the inflamed tissues[38]. Similar to the function of IL-1β, IL-6 also plays a role by amplifying the innate immune response by recruiting additional immune mediators[39]. Clinical research found that the expression of IL-1β and IL-6 in serum and alveolar lavage fluid are significantly increased in patients infected with SARS-CoV-2[5]. Our results also demonstrate that the levels of IL-1β and IL-6 in the sera and lung are increased by the lung-specific expression of N protein, but the expressions of IL-1β and IL-6 are significantly decreased by the administration of MCC950, Ac-YVAD-cmk, or in AAV-N-infected NLRP3$^{-/-}$ mice. More interestingly, in the sepsis mouse model[40], N protein significantly increases the time of mouse death, whereas the addition of MCC950 or Ac-YVAD-cmk not only delays the time of death but also improves the survival rate of the mice. This suggests that SARS-CoV-2 N protein may cause lung injury through inducing the production of proinflammatory factors such as IL-1β and IL-6.

NLRP3 inflammasome is composed of a sensor protein (NLRP3), an adaptor protein (ASC), and an effector protein (Caspase-1)[15]. During the activation of NLRP3 inflammasome, NLRP3 deubiquitination[41] and self-aggregation[27] first occurs,

followed by ASC recruitment and oligomerization[28], and then pro-caspase-1 is cut into active caspase-1 (p20), which can cleave pro-IL-1β and pro-IL-18 to produce mature IL-1β (p17) and IL-18, respectively[16]. Our results prove that SARS-CoV-2 N protein can directly interact with NLRP3 to promote the aggregation of NLRP3 protein and further studies indicate that N protein can accelerate the interaction between NLRP3 and ASC, thus facilitating the oligomerization of ASC. Immunofluorescence experiments prove that the N-NLRP3-ASC can form a complex, leading to the activation of NLRP3 inflammasome. Even more interestingly, a Co-IP experiment indicates that the CTD domain of N protein (260aa–340aa)[33] mediates the interaction with NLRP3. Moreover, missing this domain, the levels of p17, p20, and ASC oligomerization is significantly reduced. This further indicates that the interaction between N and NLRP3 protein is essential for activating the NLRP3 inflammasome. It is worth noting that our results show that N protein has no effect on the expression of IL-18, which also as an effector protein downstream of the NLRP3 inflammasome[15], seeming a little contradiction. We speculate that maybe N protein had no effect on the mRNA level of IL-18, leading to an extremely low level of protein expression, so the effect of N protein on IL-18 is not detected in our experiment.

In conclusion, our study demonstrate that SARS-CoV-2 nucleocapsid protein (N) can directly interact with NLRP3 to promote the assembly and activation of NLRP3 inflammasome, thus leading to the production of a large number of inflammatory factors and the occurrence of lung injury in mice. Our data indicate that inhibition of NLRP3 inflammasome can reduce "cytokine storm" and lung injury caused by SARS-CoV-2 infection, suggesting that NLRP3 inflammasome is a potential target for COVID-19 therapy.

## Methods
**Mice.** Wild-type C57BL/6 mice were purchased from Guangdong Medical Laboratory Animal Center, Guangzhou, China. *NLRP3*$^{-/-}$ C57BL/6 mice have been bred in our laboratory[14]. Mice were bred and maintained under specific pathogen-free conditions. Six-week-old mice were tail vein-injected with 300 μl of AAV-Lung-EGFP-N ($5 \times 10^{11}$ vg) or AAV-Lung-EGFP (purchased from OBiO

Technology, Shanghai, China). Sera were collected at 1 week and 3 weeks for all groups from retro-orbital. After 4 weeks, mice were treated with LPS (30 mg/kg) by intraperitoneal injection[40]. Mice were killed and tissues were collected for histoimmunofluorescence or histopathology analysis.

Mouse BMDMs were isolated from 6-week-old male C57BL/6 mice. Cells were cultured in RPMI 1640 medium with 10% fetal bovine serum and 10% GM-CSF conditioned medium from L929 cells for 6 days.

**Ethics statement**. All animal studies were performed in accordance with the principles described by the Animal Welfare Act and the National Institutes of Health Guidelines for the care and use of laboratory animals in biomedical research. All procedures involving mice and experimental protocols were approved by Institute of Laboratory Animal Science, Jinan University. The animal ethic committee number is 20200828-09.

**Cell lines and cultures**. Embryonic kidney cell lines (HEK293T), human pulmonary epithelial cell (A549), and human monocytic cell lines (THP-1) were purchased from American Type Culture Collection. HEK293T and A549 cells were grown in Dulbecco's modified Eagle medium and THP-1 cells were grown in RPMI 1640 medium. All the medium was supplemented with 10% fetal calf serum, 100 U/ml penicillin, and 100 μg/ml streptomycin, and maintained at 37 °C in a 5% CO$_2$ incubator. THP-1 cells were differentiated into macrophages by stimulating with PMA for 12 h. The cells were then cultured in fresh medium for 24 h and stimulated with Nigericin, LPS, or ATP. Supernatants were collected for measurement of IL-6, IL-18, and IL-1β (p17) proteins. Cells were collected for qRT-PCR and immunoblot analysis.

**Reagents and antibodies**. Human IL-1β ELISA kit (557966) was purchased from BD Biosciences. Human IL-6 (EK106/2-96), human IL-18 (EK118-48), mouse IL-6 (EK206/3-96), mouse IL-1β (EK201B/3-96), and mouse IL-18 (EK218-96) ELISA kit were purchased from Multi Sciences. PMA (880134P), Nigericin (481990), LPS (L2630), and ATP (A1852) were purchased from Sigma-Aldrich. Lipofectamine 2000 reagent (11668019) and dansylsarcosine piperidinium salt (DSS) (A39267) were purchased from Invitrogen. Trizol reagent (15596018) was purchased from Ambion. MCC950 (S7809) was purchased from Selleck. Ac-YVAD-cmk (16E02-MM) was purchased from Invivogen.

Anti-NLRP3 (D4d8T, 1 : 1000), anti-Caspase-1 (D7F10, 1 : 1000), and anti-IL-1β (D3U3E, 1 : 1000) antibodies were purchased from Cell Signaling Technology. Anti-NLRP3 (ALX-804-818, 1 : 200) was purchased from Enzo Life Science. Anti-SARS-CoV-2-N (A20021, 1 : 1000), anti-NEK7 (A19816, 1 : 1000), and anti-β-actin (AC026, 1 : 2000) antibody were purchased from ABclonal. Anti-Flag (F3165, 1 : 1000), anti-HA (H6908, 1 : 2000), and anti-GAPDH (G8759, 1 : 2000) were purchased from Sigma. Anti-ASC (sc-271054, 1 : 500) was purchased from Santa Cruz Biotechnology. Rabbit IgG (PA1-28573, 1 : 5000) and Mouse IgG (31464, 1 : 5000) were purchased from Invitrogen. Anti-mouse/rabbit IgG Dylight 649 (A23620, 1 : 200), anti-mouse/rabbit IgG Dylight cy3 (A22210, 1 : 200), and anti-mouse/rabbit IgG fluorescein isothiocyanate (FITC) (A22110, 1 : 200) were purchased from Abbkine.

**RNA extract and qRT-PCR**. Total RNAs were extracted by using Trizol reagent (Invitrogen, CA) according to the manufacturer's instructions; the RNA was then reverse-transcribed into cDNA at 42 °C for 60 min followed by 72 °C for 10 min with reserve transcriptase. Real-time qRT-PCR was performed using the Roche LC480 and SYBR RT-PCR Kits (Roche, 52427220). The real-time PCR primers were designed by Primer-blast, NCBI (www.ncbi.nlm.nih.gov), and their sequences are listed in Supplementary Table 1.

**Plasmid construction**. The eight genes (N, M, E, 3a, 6, 7a, 8, and 10) of SARS-CoV-2 (GenBank accession number MN908947.3) were synthesized by GenScript, Nanjing, China, and cloned to pcDNA3.1(+) expression vector with HA-tag or Flag-tag. For the truncated forms of SARS-CoV-2-N, the PCR productions were inserted into the XbaI and BamHI sites of plasmid pcDNA3.1(+)–3× flag. The sequences of primers were shown in Supplementary Table 2. Plasmids pcDNA3.1 (+)–3× flag-NLRP3/Caspase-1/ASC/IL-1β and pCAGGS-HA-NLRP3/ASC/Caspase-1 were constructed previously by our laboratory.

**Lentivirus-mediated gene expression and transfection**. HEK293T cells were seeded in 10 cm dishes and transfected with plenti-3× flag-SARS-CoV-2-N or negative control empty vector along with the packing plasmids psPAX2 and pMD2G using Lipo2000 transfection reagent. The medium was changed 12 h after transfection. Thirty-six hours and 60 h post transfection, cell supernatants containing lentivirus were collected and filtrated with 0.45 μm filter. PMA-differentiated THP-1 macrophages and GM-CSF-differentiated BMDMs were introduced with lentivirus for 24 h adding polybrene (Sigma, TR-1003). At 48 h post introduction, the cells were selected with puromycin (Sigma, P8833) for 4–7 days. HEK293T cells were transfected with indicated plasmids using PEI (Sigma, T0956) and THP-1 or A549 cells were transfected with indicated plasmids using Lipo2000 transfection reagent. The system of the NLRP3 inflammasome in

HEK293T cells in vitro, please refer to previous papers published by our lab[14]. Briefly, HEK293T cells were treated with NLRP3 (60 ng), pro-Caspase-1 (10 ng), ASC (10 ng), or pro-IL-1β (300 ng), and then transfected with plasmids expressing indicated proteins for 48 h. Cell supernatants were collected and analyzed for mature IL-1β level by ELISA assay, and cells were lysed and proteins in the cell analyzed by western blotting.

**Enzyme-linked immunosorbent assay**. The concentration of culture cell supernatants and mice serum of IL-6, IL-1β, and IL-18 proteins were measured by related ELISA Kit according to manufacturer's instructions. Briefly, 100 μl sample was added to ELISA well, incubated 2 h at room temperature (RT), and then aspirated and washed five times. Detected antibody (100 μl) was added to each well, incubated for 1 h at RT, and aspirated and wash five times. Enzyme working reagent (100 μl) was added to each well, incubated 30 min at RT, and aspirated and wash seven times. 3,3',5,5'tetramethylbenzidine (TMB) one-step substrate reagent (100 μl) was added to each well, incubated 30 min at RT. Stop solution (50 μl) was added to each well. Read absorbance at 450 nm within 30 min, wavelength correction at 570 nm.

**Western blotting**. PMA-differentiated THP-1 macrophages and GM-CSF-differentiated BMDMs were washed twice with PBS and dissolved in lysis buffer (50 mM Tris-HCl pH 7.4, 150 mM NaCl, 0.1% Nonidet-p40, 5 mM EDTA, and 10% glycerol). A549 cells and HEK293T cells were prepared in lysis buffer (50 mM Tris-HCl pH 7.4, 300 mM NaCl, 1% Triton X-100, 5 mM EDTA, and 10% glycerol). Protease inhibitor (10%, Roche, 04693116001) was added to lysis buffer before using. Protein concentration was measured by Bradford assay (Bio-Rad, Richmond, CA). Cell lysates (50 μg) were electrophoresed on 8−12% SDS-polyacrylamide gel electrophoresis (PAGE) and transferred to nitrocellulose membranes (Amersham, Piscataway, NJ). Nonspecific bands of Nitrocellulose membrane (NC) membranes were blocked by 5% skim milk for 2 h. NC membranes were washed three times with PBS with 0.1% Tween and incubated with indicated antibodies. Protein bands were visualized using a Bio-Rad Image Analyzer (Serial number 733BR3722).

**Co-immunoprecipitation assays**. A549, HEK293T, PMA-differentiated THP-1 cells, and THP-1-SARS-CoV-2-N stable cells grown to 70–80% confluence in 6 cm/10 cm dishes were co-transfected with indicated plasmids for 24–48 h. Transfected HEK293T cells or A549 cells were lysed in lysis buffer (50 mM Tris-HCl pH 7.4, 300 mM NaCl, 1% Triton X-100, 5 mM EDTA, and 10% glycerol) and transfected THP-1 cells were dissolved in lysis buffer (50 mM Tris-HCl pH 7.4, 150 mM NaCl, 0.1% Nonidet-p40, 5 mM EDTA, and 10% glycerol). The lysates were rotated at 4 °C for 30 min and centrifuged at 13,500 × g for 10 min to remove debris. An aliquot of supernatants was used as Input, and the rest of the supernatants were incubated with indicated antibodies overnight at 4 °C and mixed with Protein G Sepharose beads (GE Healthcare, Milwaukee, WI, USA) for 2 h at 4 °C. Immunoprecipitates were washed four to six times with respective lysis buffer, boiled in protein loading buffer, and analyzed by western blotting.

**His pull-down assays**. To construct pET-28a-His-N, the N gene was sub-cloned into pET-28a-His at EcoRI and XhoI sites. Plasmid pET-28a-His-N was transfected into Escherichia coli strain BL21. After growing in Kanamycin-resistant Lysogeny Broth (LB) medium at 37 °C until OD$_{600}$ reached 0.6–0.8, isopropyl β-D-1-thiogalactopyranoside was added to a concentration of 0.2 mM and medium was transferred to 16 °C for 12–16 h. Cells were collected and sonicated in lysis buffer (50 mM NaH$_2$PO$_4$, 100 mM phenylmethylsulfonyl fluoride (PMSF), 300 mM NaCl, 10 mM imidazole pH 8.0). Lysates were centrifuged at 13,500 × g for 15 min to move debris. Supernatants were loaded into Ni-NTA Agarose columns (QIAGEN, 30210), shaken at 4 °C for 2 h, and slowly flowed from the columns and washed twice with wash buffer (50 mM NaH$_2$PO$_4$, 100 mM PMSF, 300 mM NaCl, 20 mM imidazole pH 8.0). Recombinant His-N protein was eluted using elution buffer (50 mM NaH$_2$PO$_4$, 100 mM PMSF, 300 mM NaCl, and 250 mM imidazole pH 8.0). Recombinant His-N containing reduced imidazole was replaced into PBS using Millipore ultrafiltration tube. Ni-NTA Agarose were incubated with purified His-N at 4 °C for 2 h and incubated with cell lysates prepared from HEK293T cells transfected with pcDNA3.1(+)–3× flag-NLRP3 for 16 h at 4 °C. Precipitates were washed four times with PBS, boiled in protein loading buffer, and separated by SDS-PAGE.

**Confocal microscopy**. HEK293T and A549 cells grown on sterile cover slips were transfected with indicated plasmids at 40% confluence for 24 h. PMA-differentiated THP-1-CT or THP-1-N stable cells were grown on sterile cover slips for 24 h. Cells were fixed with 4% paraformaldehyde for 30 min, washed three times with ice-cold PBS containing 0.1% bovine serum albumin (BSA), permeabilized with PBS containing 0.2% Triton X-100 for 5 min, washed three times with PBS, and finally blocked with PBS containing 5% BSA for 1 h. Cells were incubated overnight with anti-HA antibody, anti-Flag antibody, and anti-NLRP3 antibody (1 : 200 in PBS) followed by staining with Cy3-conjugated donkey anti-rabbit IgG, FITC-conjugated donkey anti-mouse IgG, and Daylight 649-conjugated donkey anti-rabbit IgG secondary antibody (Abbkine) (1 : 100 in wash buffer). Nuclei were stained with DAPI (4′,6-diamidino-2-phenylindole) for 5 min, and the cells were washed three times. Cells were viewed under confocal fluorescence microscope (Leica, TCS, SP8).

**ASC oligomerization analysis**. HEK293T cells, PMA-differentiated THP-1 cells, and GM-CSF-differentiated BMDMs were lysed in lysis buffer, gently shaken at 4 °C for 30 min, and centrifuged at 3380 × g at 4 °C for 10 min. Pellets were washed three times with PBS and re-suspended in 200 µl PBS. DSS (2 mM) was added to re-suspended pellets, which were cross-linked at 37 °C for 30 min. The samples were centrifuged at 3380 × g for 10 min, cross-linked pellets were re-suspended in 50 µl 2× SDS loading buffer, boiled for 10 min, and analyzed by western blotting.

**Statistical analyses**. All experiments were repeated at least three times with similar results. Statistical analysis was carried out using the $t$-test for two groups and one-way analysis of variance for multiple groups (GraphPad Prism7). The date was considered statistically significant when $*P \leq 0.05$, $**P \leq 0.01$, $***P \leq 0.001$; ns stands for not significant.

**Reporting summary**. Further information on research design is available in the Nature Research Reporting Summary linked to this article.

## Data availability

RNA-Seq data accessed are deposited in the GEO database under the accession code GSE155106. All other data are included in the article and Supplemental Information, or available from the authors upon reasonable requests. Source data are provided with this paper.

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

## Acknowledgements

This work was supported by National Natural Science Foundation of China (81730061), Guangdong Province "Pearl River Talent Plan" Innovation and Entrepreneurship Team Project (2017ZT07Y580), China Postdoctoral Science Foundation (2020M683177 and 2020T130046ZX), Natural Science Foundation of Guangdong Province general program (2020A1515010369), and Fundamental Research Funds for the Central Universities (21620401 and 11620301). Science & Technology Planning Project of Guangdong Province Office of Education (2018KCXTD007, 2021XK16), Special project for the prevention and control of new crown pneumonia in Guangdong colleges and universities (2020KZDZX1060).

## Author contributions

P.P., M.S., Z.Y., G.L., Y.L., and J.W. contributed to the design of experiments. P.P., M.S., Z.Y., W.G. K.C., M.T., F.X., Z.W., J.W., Y.J., W.W., P.W., J.Z., W.C., Z.L., X.C., Z.L., Q.Z., M.X., G. L., and Y.L. contributed to the conduction of experiments. P.P., M.S., Z.Y., W.G., K.C., M.T., F.X., Z.W., J.W., Y.J., W.W., P.W., J.Z., W.C., Z.L., X.C., Z, Q.Z., M.X., G.L., Y.L., and Jianguo Wu contributed to the reagents. P.P., M.S., Z.Y., Y.L., and J.W. contributed to writing of the paper. P.P., G.L., Y.L., and J.W. contributed to editing of the paper.

## Competing interests

The authors declare no competing interests.
