## [Peer Review File · Nature Communications]

REVIEWER COMMENTS

Reviewer #1 (Remarks to the Author):

The emergence and significant global health impact of SARS CoV2 combined with the lack of effective therapies to treat the pulmonary inflammation associated with infections has highlighted the critical need to identify the mechanisms and effectors of how the host immune system is triggered. In this study, the authors have identified the nucleocapsid (N) protein of SARS CoV2 as an activator of the NLRP3 inflammasome through a range of in vitro and in vivo models of activation and inflammation. The authors identified that ectopically expressed N protein interacts with and drives NLRP3 activation, inducing IL-1 β maturation and induction of the subsequent inflammatory response, inducing increased IL-6 production which has been identified as associated with poor prognostic outcomes in infected patients.

Understandably, the study is based on ectopic overexpression of the individual SARS CoV2 proteins and pseudo models of NLRP3 inflammasome biology/function. However, while N protein is identified in Figures 1 and 2 as the potential inflammatory inducer moving forward, there is concern that the gross overexpression levels of N protein compared to the other SARS CoV2 protein may induce a stress response and activation of NF- κ B which is responsible for increased pro-IL-1 β levels. Indeed, given the difference identified by the authors in IL-1 β and IL-18 responses, the levels of both pro-proteins should be examined in their study to also address the potential role of N protein in simply inducing NF- κ B activation and subsequent gene transcription which would explain these early results. Furthermore, it could be argued that given the considerably lower expression of the other proteins (ie. Fig 1C) compared to N protein, particularly the non-structural proteins, their induction of IL-1 β in Figs 1d and e are superior to that of N protein. Therefore, why was N protein selected for further analysis as opposed to others? Many of the experiments conducted also would have been better controlled given the preference for N protein, to have expressed one of the other SARS CoV2 proteins to demonstrate the specificity of effect rather than the empty vector.

A major concern is also the exclusive use of Ac-YVAD-cmk and its description as an NLRP3 inflammasome inhibitor. It is a caspase-1 inhibitor and the experiments would have been better served to have used the specific NLRP3 inhibitor MCC950 both in vitro and in vivo if conclusions such as those on line 315 that 'Ac-YVAD-cmk, a specific inhibitor of NLRP3 inflammasome'. Using a caspase-1 inhibitor in the in vivo experiments of Fig 7g also confounds as LPS itself can activate the NLRP3 inhibitor, therefore this experiment should also include treating AAV-CT/LPS challenged mice (again the empty vector rather than a comparative expressed protein) with YVAD. There are also inconsistencies with the confocal microscopy. While the cell biology activation studies expressed the vectors for 48h in cells, imaging was conducted after 24h. Are there differences in tagged protein expression between 24 and 48h? There appears a different expression/localization 'phenotype' when ASC was expressed for 48h (Fig 6i) as opposed to the other NLRP3 and/or EGFP-N+ for 24h. The authors also state that NLRP3 colocalizes with N protein and forms specks in Figure 6a, but this is not the case with A549 cells. The authors also state that ASC and N protein do not colocalize, however it appears in Fig 3h, i that ASC and N protein are not expressed in the same cells, but in the cells that both are expressed, there does appear to be colocalization (ie Fig 3h, bottom right panel- left side of panel cells of colocalization). Proteins also appear to be extensively expressed at high levels and diffuse throughout cytosol, the resolution of these figures makes it hard to distinguish the specificity of the colocalization given the high expression of both proteins. Ectopic expression of NLRP3 can activate the complex in transfected cells (as evidenced by IL-1 β production in mock transfected NLRP3 reconstituted HEK293T cells in Figs 1 and 2). Why is this not evident in Figure 3? Supplementary data should be included providing widefield representation of speck formation (or not in the case of controls) and quantified. These studies should also be complemented with expression of a non-activating/interacting SARS CoV2 protein.

Overall, the study is important in identifying potential SARS CoV2 activators of the NLRP3 inflammasome leading to induction of inflammation. Identifying a role for the inflammasome in SARS CoV pathology would provide potential therapeutic strategies to reduce the inflammation that is associated with poor clinical outcomes.

Figure 5 is stated as (line 219) 'N and ASC could interact with each other in the presence of NLRP3...' which contradicts the statement regarding confocal colocalization on line 195.

Figure 7: what is the statistical analysis method used to determine statistical significance- this is

not mentioned in the figure legend and given there are overlapping error bars and presented as SEM, this is important to determine how significance was determined? Also, the figure legends imply there should be 8 mice per group, yet only 6 data points are included in the figures? ORF7b is not mentioned as a SARS CoV2 protein in the introduction.

Reviewer #2 (Remarks to the Author):

In this manuscript the authors demonstrate that N protein from SARS-CoV-2 induced inflammatory responses and facilitates the production of proinflammatory cytokines such as IL-6 and IL-1 β . Mechanistically, the authors claim that N protein interacts with NLRP3 directly, promoting direct binding of NLRP3 and ASC leading to activation of the inflammasome.

1. It is absolutely incorrect to say that "the effect of the NLRP3 inflammasome on regulating SARS-CoV-2-induced inflammatory responses has not been reported". See PMID: 33231615 and PMID: 33079210, these papers clearly demonstrate NLRP3 activation in COVID-19 patients and in response to SARS-CoV-2 infection and should be acknowledged. Similar sentences are also found elsewhere in the manuscript (lines 71-73, etc). The authors don't need, and actually should not, neglect previous studies to "sell" their story. This study is nice and relevant because it is the first to propose a mechanism by which SARS-CoV-2 N protein trigger the NLRP3 inflammasome. This should be sufficient novelty to support this paper.

2. The authors use Ac-YVAD-cmk, to inhibit NLRP3 inflammasome in mouse models treated with N protein. However, Ac-YVAD-cmk can inhibit inflammatory caspases in general (Casp1, Casp11, casp12, etc.). The authors should reproduce this finding using *Nlrp3*^{-/-} mice, which is readily available in Jax and many labs. Experiments with more specific NLRP3 inhibitors such as MCC950 would also be useful.

3. In certain conditions, NEK7 is important to license NLRP3 activation through direct binding or NLRP3 (PMID: 31189953). Is NEK7 required for N protein mediated NLRP3-ASC interaction and inflammasome activation?

4. If N protein is related to inflammasome activation, how the authors explain the increased production non-inflammasome cytokines such as IL-6, TNF- α , etc? Are there multiple mechanisms by which N protein promotes inflammation? Is it possible that N protein mediates a global induction of transcriptional responses? This would explain the increased inflammasome activation and general cytokines and chemokines. The authors should measure the expression of *Nlrp3* (RNA and protein) to evaluate if the reported N protein - mediated NLRP3 inflammasome activation is transcriptional regulated or posttranslational regulated via binding of NLRP3 to ASC.

5. Multiple inflammasomes uses ASC to trigger caspase-1 activation and inflammatory responses (AIM2, NLRC4, etc.). Is the N protein mediated inflammasome activation specific to NLRP3? Or N protein triggers additional ASC-dependent inflammasomes?

Minor:

1- Does the authors have a hypothesis to explain how a protein that is part of the envelop (therefore it is not internalized during viral entry), reaches the cytoplasm to trigger NLRP3 activation? This should be discussed.

2. Line 59: What the authors want so say with "the NLRP3 inflammasome is closely related to RNA virus infection"? This sentence should be adjusted.

3. Protein E from SARS-CoV1 is known to be porin that transports K⁺ ions and activates the NLRP3 inflammasome. Given the high homology of E from CoV-1 and Cov-2, it would be expected that E

from CoV-2 triggers inflammasome. But this was not the case according to figure 1D. The authors should discuss that.

4. Lines 458, 460, 461, 495, 497: use the symbol "o". It is not properly configured.

5. In figure 2, the authors show that N protein induced IL-1B (p17) cleavage and Casp-1 (p20) maturation in a dose dependent manner (Fig2b, c). However, the specific concentration of N protein that induces these phenotypes in a dose dependent manner is not clear (Fig2b)? Supplemental graphics of densitometry analysis would be useful to better visualize these data.

6. At Fig. 6, the legend should describe which means the solid, dashed arrows.

7. IL-18 and Inflammasome activation have been identified as biomarkers for mortality after SARS CoV-2 infection in patients (Rodrigues et al., JEM 2020; Lucas et al., Nature 2020). However, in the present study, the authors did not observe N protein effects on the IL-18 expression in mouse. Did the authors measure IL-18 expression in the mice sera? The authors should at least discuss these findings in the light of previous publications.

Responses to the Reviews' Comments

REVIEWER COMMENTS

Reviewer #1 (Remarks to the Author):

The emergence and significant global health impact of SARS CoV2 combined with the lack of effective therapies to treat the pulmonary inflammation associated with infections has highlighted the critical need to identify the mechanisms and effectors of how the host immune system is triggered. In this study, the authors have identified the nucleocapsid (N) protein of SARS CoV2 as an activator of the NLPR3 inflammasome through a range of in vitro and in vivo models of activation and inflammation. The authors identified that ectopically expressed N protein interacts with and drives NLPR3 activation, inducing IL-1b maturation and induction of the subsequent inflammatory response, inducing increased IL-6 production which has been identified as associated with poor prognostic outcomes in infected patients.

Authors' Response: Thank you very much for your remarks.

Reviewer #1' Comment: Understandably, the study is based on ectopic overexpression of the individual SARS CoV2 proteins and pseudo models of NLPR3 inflammasome biology/function. However, while N protein is identified in Figures 1 and 2 as the potential inflammatory inducer moving forward, there is concern that the gross overexpression levels of N protein compared to the other SARS CoV2 protein may induce a stress response and activation of NF-kB which is responsible for increased pro-IL-1b levels. Indeed, given the difference identified by the authors in IL-1b and IL-18 responses, the levels of both pro-proteins should be examined in there study to also address the potential role of N protein in simply inducing NF-kB activation and subsequent gene transcription which would explain these early results. Furthermore, it could be argued that given the considerably lower expression of the other proteins (ie. Fig 1C) compared to N protein, particularly the non-structural proteins, their induction of IL-1b in Figs 1d and e are superior to that of N protein. Therefore, why was N protein selected for further analysis as opposed to others?

Authors' Response: Thank you for your comments.

According to your suggestion, we have performed additional experiments to determine the role of N protein in the regulation of NF- κ B signaling pathway in the revised manuscript. Notably, the levels of NLRP3 mRNA (Revised Fig. S1c, top) as well as NLRP3, pro-IL-1 β , and pro-IL-18 proteins (Revised Fig. S1c, bottom) were relatively unaffected by N protein. These results implicated that N protein regulates NLRP3, pro-IL-1 β , and pro-IL-18 independent on NF- κ B signaling pathway. We noticed that IL-1 β protein promoted the production of inflammatory factors and cytokines, including IL-6, TNF- α , CXCL10, IL-11, IL-13, and CCL2 (Revised Fig. S1b). Thus, we speculate that N protein promotes NLRP3 inflammasome activation, thereby inducing mature IL-1 β secretion, which subsequently plays a key role in the induction of inflammatory factors and cytokines.

In our results, we found that N protein was the highest expression compared with other proteins, including M, E, 3a, 6, 7a, 8 and 10 (Revised Fig. 1c). To investigate this problem, THP-1 cells (Revised Fig. S1a) and HEK293T cells (Revised Fig. S1b) were transfected with indicated plasmids at the same concentration for 48 h. Total RNAs were extracted and the mRNA levels of the indicated genes were detected by qRT-PCR. We noticed that the mRNA level of each gene was high, especially N, 3a, 7a, 8, and 10 genes (Revised Fig. S1a, b), indicating that the plasmid transfections were not the issue. Thus, we speculated that on one hand, the nonstructural proteins are unstable and degrades in the cell. On the other hand, the proteins themselves are too small to be detected. In the preliminary screening, we noticed that in both constructed HEK293T-inflammasome system and THP-1 cells, the level of N-induced IL-1 β secretion was the highest among all proteins (Revised Fig. 1d, e) and thus, we choice N protein for further studies in this work.

Reviewer #1' Comment: Many of the experiments conducted also would have been better controlled given the preference for N protein, to have expressed one of the other SARS CoV2 proteins to demonstrate the specificity of effect rather than the empty vector.

Authors' Response: Thank you for the suggestion.

According to your suggestion, we have performed several additional experiments by using SARS-CoV-2 3a protein as a control to demonstrate the specific effect of N protein on the induction of IL-1 β and NLRP3 inflammasome activation in the revised manuscript.

The new results showed that: (1) IL-1 β protein was significantly induced by the N protein, but relatively affected by the 3a protein (Revised Fig. S2). (2) The N protein could interact with the

NLRP3 protein, but the 3a protein failed to interact with the NLRP3 protein, in both HEK293T (Revised Fig. S3a) and A549 cells (Revised Fig. S3b). (3) The interaction of NLRP3 and ASC was enhanced by N protein in dose-dependent manners in HEK293T cells (Revised Fig. 5f) and A549 cells (Revised Fig. 5g), but 3a protein had no effect on the interaction of NLRP3 and ASC (Revised Fig. S4a, b). (4) N protein significantly induced ASC oligomerization (Revised Fig. 6f, g and h), but 3a protein had no effect on ASC oligomerization (Revised Fig. S5a). (5) In the presence of N protein, NLRP3 co-localized with N and formed specks; however, in the presence of 3a protein, NLRP3 co-localized with 3a, but failed to form specks (Revised Fig. 6a). (6) When N, ASC, and NLRP3 expressed together, the three proteins were co-localized to form “sphere-like” structures, however, in the presence of 3a, ASC, and NLRP3 together, the three proteins were co-localized, but failed to form “sphere-like” structures (Fig. 6i, j). Taken together, these results demonstrated that SARS-CoV-2 N protein, but not the viral 3a protein, induces IL-1 β protein and promotes the assemble of the NLRP3 inflammasome complex.

Reviewer #1' Comment: A major concern is also the exclusive use of Ac-YVAD-cmk and its description as an NLRP3 inflammasome inhibitor. It is a caspase-1 inhibitor and the experiments would have been better served to have used the specific NLRP3 inhibitor MCC950 both in vitro and in vivo if conclusions such as those on line 315 that ‘Ac-YVAD-cmk, a specific inhibitor of NLRP3 inflammasome’. Using a caspase-1 inhibitor in the in vivo experiments of Fig 7g also confounds as LPS itself can activate the NLRP3 inhibitor, therefore this experiment should also include treating AAV-CT/LPS challenged mice (again the empty vector rather than a comparative expressed protein) with YVAD.

Authors' Response: Thank you for your comments.

As you indicated, we have changed the description “Ac-YVAD-cmk, an inhibitor of the NLRP3 inflammasome” to “Ac-YVAD-cmk, an inhibitor of the Caspase-1” in the revised manuscript.

More importantly, according to your suggestion, we have supplemented several in vitro and in vivo experiments by using the specific NLRP3 inhibitor MCC950 in the revised manuscript. The new results indicated that: (1) Lentivirus-N-mediated inductions of inflammatory factors (Revised Fig. 1g) or mature IL-1 β (Revised Fig. S6e) were repressed by MCC950. (2) AAV-Lung-N-induced expression of IL-1 β , IL-6, TNF- α , CXCL10, CCL2, IL-11, IL-7, and CXCL8 were significantly

suppressed by MCC950 and Ac-YVAD-cmk (Revised Fig. 1i). (3) The protein levels of IL-1 β and IL-6 were much higher in the sera of AAV-N-infected mice as compared to AAV-CT-infected mice, however, AAV-N-mediated inductions of IL-1 β and IL-6 were repressed by both Ac-YVAD-cmk (Revised Fig. 7c, d) and MCC950 (Revised Fig. 7e, f). (4) Immunohistochemical fluorescence analyses showed that IL-1 β protein and IL-6 protein were highly expressed in the lungs of mice carrying AAV-Lung-N, whereas such activation was attenuated by Ac-YVAD-cmk and MCC950 (Revised Fig. 7g, h). (5) Hematoxylin & eosin staining analyses indicated that inflammatory lesions and tissue injuries were obvious in the lungs of mice carrying AAV-Lung-N, but such pathological changes were repressed by the administration of Ac-YVAD-cmk and MCC950 (Fig. 7i). (6) We noticed that the IL-18 protein was not affected by AAV-N in the sera of both NLRP3^{+/+} and NLRP3^{-/-} mice (Revised Fig. S6d) or by MCC950 (Revised Fig. S6c). (7) The survival rates were reduced in mice treated with LPS plus injected with AAV-Lung-N, and recovered in mice treated with LPS plus injected with AAV-Lung-N and treated with MCC950 (Revised Fig. 7l) or Ac-YVAD-cmk (Revised Fig. S7i). We added the group that include treating AAV-GFP/LPS challenged mice with MCC950 in the new experiment. Compared with AAV-GFP/LPS, the survival rates were recovered in mice treated AAV-GFP/LPS with MCC950 (Revised Fig. 7l).

Collectively, these results confirmed that the NLRP3 inflammasome is required for the induction of inflammatory response and the development of lung injury mediated by the N protein.

Reviewer #1' Comment: There are also inconsistencies with the confocal microscopy. While the cell biology activation studies expressed the vectors for 48h in cells, imaging was conducted after 24h. Are there differences in tagged protein expression between 24 and 48h? There appears a different expression/localization 'phenotype' when ASC was expressed for 48h (Fig 6i) as opposed to the other NLRP3 and/or EGFP-N+ for 24h. The authors also state that NLRP3 colocalizes with N protein and forms specks in Figure 6a, but this is not the case with A549 cells. The authors also state that ASC and N protein do not colocalize, however it appears in Fig 3h, i that ASC and N protein are not expressed in the same cells, but in the cells that both are expressed, there does appear to be colocalization (ie Fig 3h, bottom right panel- left side of panel cells of colocalization). Proteins also appear to be extensively expressed at high levels and diffuse throughout cytosol, the resolution of these figures makes it hard to distinguish the specificity of the colocalization given the high

expression of both proteins. Ectopic expression of NLRP3 can activate the complex in transfected cells (as evidenced by IL-1 β production in mock transfected NLRP3 reconstituted HEK293T cells in Figs 1 and 2). Why is this not evident in Figure 3? Supplementary data should be included providing widefield representation of speck formation (or not in the case of controls) and quantified. These studies should also be complemented with expression of a non-activating/interacting SARS CoV2 protein.

Authors' Response: Thanks for the comments and questions.

Our responses to each of your comments, respectively, are described as following.

Reviewer #1' Comment: While the cell biology activation studies expressed the vectors for 48h in cells, imaging was conducted after 24h. Are there differences in tagged protein expression between 24 and 48h?

Authors' Response: Sorry for the confusions, we did not describe clearly the conditions used for the cell biology activation studies and the immunofluorescence imaging studies in the original manuscript.

For cell biology activation studies, a reconstructed NLRP3 inflammasome system was used, in which HEK293T cells were co-transfected with plasmids encoding the four components NLRP3, ASC, pro-Casp-1 and pro-IL-1 β at lower concentrations. We noticed that IL-1 β secretion and transfected protein expression were detected at 24 h and increased to a higher level at 48 h in this system and thus, we selected 48 h for cell biology activation studies.

For the immunofluorescence imaging studies, HEK293T cells and A549 cells were co-transfected with plasmids at relatively high concentrations. We noticed that higher levels of proteins were detected at 24 h, and thus, we selected 24 h for immunofluorescence studies.

The reason for us to select the detection times was mainly due to the different concentrations of transfected plasmids.

Reviewer #1' Comment: There appears a different expression/localization 'phenotype' when ASC was expressed for 48h (Fig 6i) as opposed to the other NLRP3 and/or EGFP-N+ for 24h. The authors also state that NLRP3 colocalizes with N protein and forms specks in Figure 6a, but this is not the case with A549 cells. The authors also state that ASC and N protein do not colocalize, however it

appears in Fig 3h, i that ASC and N protein are not expressed in the same cells, but in the cells that both are expressed, there does appear to be colocalization (ie Fig 3h, bottom right panel- left side of panel cells of colocalization). Proteins also appear to be extensively expressed at high levels and diffuse throughout cytosol, the resolution of these figures makes it hard to distinguish the specificity of the colocalization given the high expression of both proteins.

Authors' Response: Sorry for the confusion, we should indicate it clearer that ASC protein was detected at 24 h, but not at 48 h (Original Fig. 6i).

To address your comment and make the results clearer, we have repeated the experiments in the revised manuscript. The new data showed that NLRP3 protein co-localized with N protein and forms specks in A549 cells (Revised Fig. 6a).

Moreover, we have provided 3D pictures (Movies) for the results in the revised manuscript. Notably, the new data showed that NLRP3 co-localized with N (Supplement Movie 1), but ASC failed to co-localize with N (Supplement Movies 2, 3).

Reviewer #1' Comment: Ectopic expression of NLRP3 can activate the complex in transfected cells (as evidenced by IL-1b production in mock transfected NLRP3 reconstituted HEK293T cells in Figs 1 and 2). Why is this not evident in Figure 3? Supplementary data should be included providing widefield representation of speck formation (or not in the case of controls) and quantified. These studies should also be complemented with expression of a non-activating/interacting SARS CoV2 protein.

Authors' Response: Again, sorry for the confusions, we did not describe clearly the conditions used for the cell biology activation studies (Original Fig. 1 and 2) and the immunofluorescence imaging studies (Original Fig. 3) in the original manuscript.

For cell biology activation studies, a reconstructed NLRP3 inflammasome system was used, in which HEK293T cells were co-transfected with plasmids encoding four components NLRP3, ASC, pro-Casp-1 and pro-IL-1 β . Our results demonstrated that IL-1 β secretion was significantly induced by N protein in this system (Revised Fig. 1d and Revised Fig. 2a, b).

For the immunofluorescence imaging studies, HEK293T cells and A549 cells were transfected with plasmids expressing NLRP3 and/or ASC as indicated, but not the four components were simultaneously expressed. Therefore, we could not detect the activation of NLRP3 inflammasome or

the secretion and production of IL-1 β in these cells (Revised Fig. 3).

According to your suggestion, we have performed several additional experiments by using SARS-CoV-2 3a protein (a non-activating/interacting SARS-CoV-2 protein) as a control to demonstrate the specificity of effect of N protein in the induction of IL-1 β and activation of the NLRP3 inflammasome in the revised manuscript.

Our new data showed that: (1) IL-1 β protein was significantly induced by the N protein, but relatively affected by the 3a protein (Revised Fig. S2). (2) The N protein could interact with the NLRP3 protein, but the 3a protein failed to interact with the NLRP3 protein, in both HEK293T (Revised Fig. S3a) and A549 cells (Revised Fig. S3b). (3) The interaction of NLRP3 and ASC was enhanced by N protein in dose-dependent manners in HEK293T cells (Revised Fig. 5f) and A549 cells (Revised Fig. 5g), but 3a protein had no effect on the interaction of NLRP3 and ASC (Revised Fig. S4a, b). (4) N protein significantly induced ASC oligomerization, but 3a protein had no effect on ASC oligomerization (Revised Fig. S5a). (5) In the presence of N protein, NLRP3 co-localized with N and formed specks; however, in the presence of 3a protein, NLRP3 co-localized with 3a, but failed to form specks (Revised Fig. 6a). (6) When N, ASC, and NLRP3 expressed together, the three proteins were co-localized to form “sphere-like” structures, however, in the presence of 3a, ASC, and NLRP3 together, the three proteins were co-localized, but failed to form “sphere-like” structures (Fig. 6i, j). Taken together, these results demonstrated that SARS-CoV-2 N protein, but not the viral 3a protein, induces IL-1 β protein and promotes the assemble of the NLRP3 inflammasome complex.

Reviewer #1' Comment: Overall, the study is important in identifying potential SARS CoV2 activators of the NLRP3 inflammasome leading to induction of inflammation. Identifying a role for the inflammasome in SARS CoV pathology would provide potential therapeutic strategies to reduce the inflammation that is associated with poor clinical outcomes.

Authors' Response: Thank you very much for your kindly remarks and comments.

Reviewer #1' Comment: Figure 5 is stated as (line 219) ‘N and ASC could interact with each other in the presence of NLRP3...’ which contradicts the statement regarding confocal colocalization on line 195.

Authors' Response: Thanks for the comment.

Sorry for the confusion. We should make the observations clearer.

In line 195: HEK293T cells and A549 cells were co-transfected with two proteins (N and NLRP3) or (N and ASC), confocal microscope analyses showed that N protein and NLRP3 protein were co-localized in the cytoplasm (Original Fig. 3f, g), but N protein failed to interact with ASC protein (Original Fig. 3h, i).

However, in line 219: HEK293T cells and A549 cells were co-transfected with three proteins (N, NLRP3, and ASC), the results showed that N and ASC could interact with each other in the presence of NLRP3 in HEK293T cells (Fig. 5b, c) and A549 cells (Fig. 5d, e), suggesting that N, NLRP3, and ASC three proteins together might form a complex N-NLRP3-ASC. To make it clearer, we have changed the description from “N and ASC could interact with each other in the presence of NLRP3 in HEK293T cells (Fig. 5b, c) and A549 cells (Fig. 5d, e)” to “N and ASC could immune-precipitate with each other in the presence of NLRP3 in HEK293T cells (Fig. 5b, c) and A549 cells (Fig. 5d, e)” in the revised manuscript.

Reviewer #1' Comment: Figure 7: what is the statistical analysis method used to determine statistical significance- this is not mentioned in the figure legend and given there are overlapping error bars and presented as SEM, this is important to determine how significance was determined? Also, the figure legends imply there should be 8 mice per group, yet only 6 data points are included in the figures?

Authors' Response: Good comment.

According to your suggestion, we have changed the statement “All experiments were repeated at least three times with similar results. All results were expressed as the mean \pm the standard Error of mean (SEM). Statistical analysis was carried out using the t-test for two groups and one-way ANOVA for multiple groups (GraphPad Prism7). The date was considered statistically significant when $P \leq 0.05$ (*), $P \leq 0.01$ (**), $P \leq 0.001$ (***)” to “All experiments were repeated at least three times with similar results. Statistical analysis was carried out using the t-test for two groups and one-way ANOVA for multiple groups (GraphPad Prism7). The date was considered statistically significant when $P \leq 0.05$ (*), $P \leq 0.01$ (**), $P \leq 0.001$ (***). ns stands for not significant.” in the Statistical Analyses section of the revised manuscript. 8 mice per group in Figure 7 Legends were described incorrectly, which has been corrected.

Reviewer #1' Comment: ORF7b is not mentioned as a SARS CoV2 protein in the introduction.

Authors' Response: Thanks again.

As you suggested, we have added ORF7b as one of SARS-CoV-2 proteins in the introduction of revised manuscript “In addition, the virus encodes a series of accessory proteins (ORF3a, ORF6, ORF7a, ORF7b, ORF8, and ORF10)”.

Reviewer #2 (Remarks to the Author):

In this manuscript the authors demonstrate that N protein from SARS-CoV-2 induced inflammatory responses and facilitates the production of proinflammatory cytokines such as IL-6 and IL-1 β .

Mechanistically, the authors claim that N protein interacts with NLRP3 directly, promoting direct binding of NLRP3 and ASC leading to activation of the inflammasome.

Authors' Response: Thank you for your remarks.

Reviewer #2' Comment: 1. It is absolutely incorrect to say that “the effect of the NLRP3 inflammasome on regulating SARS-CoV-2-induced inflammatory responses has not been reported”. See PMID: 33231615 and PMID: 33079210, these papers clearly demonstrate NLRP3 activation in COVID-19 patients and in response to SARS-CoV-2 infection and should be acknowledged. Similar sentences are also found elsewhere in the manuscript (lines 71-73, etc). The authors don't need, and actually should not, neglect previous studies to “sell” their story. This study is nice and relevant because it is the first to propose a mechanism by which SARS-CoV-2 N protein trigger the NLRP3 inflammasome. This should be sufficient novelty to support this paper.

Authors' Response: Thank you very much for your comments.

Actually, as the two papers (PMID: 33231615 and PMID: 33079210) have not been published when we prepared the original manuscript, we could not include the information in the Introduction section of our original manuscript. As you indicated that we don't need, and should not, neglect previous studies to “sell” our story.

In the revised manuscript, we have changed the statement “However, the role of the NLRP3 inflammasome in regulating the initiation of the cytokine storm in severe COVID-19 and the effect of SARS-COV-2 infection on triggering the activation of the NLRP3 inflammasome remained unknown.” to “Recent studies reported that inflammasomes are induced upon SARS-CoV-2 infection and associated with COVID-19 severity in patients”. The two papers have been cited in the revised manuscript (Revised Ref. 20, 21). Rodrigues et al., J Exp Med. 218(3), e20201707 (2021); Toldo et al., Inflamm Res. 70(1), 7-10 (2021).

Reviewer #2' Comment: 2. The authors use Ac-YVAD-cmk, to inhibit NLRP3 inflammasome in

mouse models treated with N protein. However, Ac-YVAD-cmk can inhibit inflammatory caspases in general (Casp1, Casp11, casp12, etc.). The authors should reproduce this finding using Nlrp3^{-/-} mice, which is readily available in Jax and many labs. Experiments with more specific NLRP3 inhibitors such as MCC950 would also be useful.

Authors' Response: Thank you very much for the suggestions.

According to your suggestion, we have carried out additional experiments by using the specific NLRP3 inhibitor MCC950 and the NLRP3^{-/-} mice in the revised manuscript.

The new results indicated that: (1) Lentivirus-N-mediated inductions of inflammatory factors (Revised Fig. 1g) or mature IL-1 β (Revised Fig. S6e) were repressed by MCC950. (2) AAV-Lung-N-induced expression of IL-1 β , IL-6, TNF- α , CXCL10, CCL2, IL-11, IL-7, and CXCL8 were significantly suppressed by MCC950 and Ac-YVAD-cmk (Revised Fig. 1i). (3) The protein levels of IL-1 β and IL-6 were much higher in the sera of AAV-N-infected mice as compared to AAV-CT-infected mice, however, AAV-N-mediated inductions of IL-1 β and IL-6 were repressed by both Ac-YVAD-cmk (Revised Fig. 7c, d), MCC950 (Revised Fig. 7e, f) and in NLRP3^{-/-} mice (Revised Fig. 7j, k). (4) Immunohistochemical fluorescence analyses showed that IL-1 β protein and IL-6 protein were highly expressed in the lungs of mice carrying AAV-Lung-N, whereas such activations were attenuated by Ac-YVAD-cmk, MCC950 (Revised Fig. 7g, h) and in NLRP3^{-/-} mice (Revised Fig. S6f, g). (5) Hematoxylin & eosin staining analyses indicated that inflammatory lesions and tissue injuries were obvious in the lungs of mice carrying AAV-Lung-N, but such pathological changes were repressed by the administration of Ac-YVAD-cmk, MCC950 (Fig. 7i) and in NLRP3^{-/-} mice (Revised Fig. S6h). (6) We noticed that the IL-18 protein was not affected by AAV-N in the sera of both NLRP3^{+/+} and NLRP3^{-/-} mice (Revised Fig. S6d) or by MCC950 (Revised Fig. S6c). (7) The survival rates were reduced in mice treated with LPS plus injected with AAV-Lung-N, and recovered in mice treated with LPS plus injected with AAV-Lung-N and treated with MCC950 (Revised Fig. 7l) or Ac-YVAD-cmk (Revised Fig. S7i).

Reviewer #2' Comment: 3. In certain conditions, NEK7 is important to license NLRP3 activation through direct binding or NLRP3 (PMID: 31189953). Is NEK7 required for N protein mediated NLRP3-ASC interaction and inflammasome activation?

Authors' Response: Thanks for the suggestion.

According to your suggestion, we have carried out addition experiments to evaluate the role of NEK7 in NLRP3 inflammasome activation mediated by SARS-CoV-2 N protein in the revised manuscript.

siRNAs specific targeting NEK7 were generated. The new results showed that NEK7 mRNA (Revised Fig. S5b–d, top) and NEK7 protein (Revised Fig. S5b–d, bottom) were significantly attenuated by si-NEK7-3. THP-1 cells stably infected with N-Lentivirus were differentiated into macrophages and then transfected with si-NEK7-3 and treated with Negericin. Notably, IL-1 β protein was induced by N protein and significantly stimulated by Negericin, whereas N-promoted and Negericin-induced IL-1 β proteins were relatively not affected by si-NEK7-3 (Revised Fig. S5e). Similarly, N-promoted ASC oligomerization in THP-1 cells (Revised Fig. S5f) as well as N-facilitated NLRP3-ASC interaction in both THP-1 cells (Revised Fig. S5g), HEK293T cells (Revised Fig. S5h) and A549 cells (Revised Fig. S5i) were not influenced by si-NEK7-3. These results indicated that NEK7 is not required for N protein-mediated NLRP3 inflammasome activation and NLRP3-ASC interaction.

We have also cited the paper (PMID: 31189953) (Sharif et al. Structural mechanism for NEK7-licensed activation of NLRP3 inflammasome. *Nature*. 570(7761), 338-343 (2019).) in the revised manuscript (Revised Ref. 34)

Reviewer #2' Comment: 4. If N protein is related to inflammasome activation, how the authors explain the increased production non-inflammasome cytokines such as IL-6, TNF-a, etc? Are there multiple mechanisms by which N protein promotes inflammation? Is it possible that N protein mediates a global induction of transcriptional responses? This would explain the increased inflammasome activation and general cytokines and chemokines. The authors should measure the expression of Nlrp3 (RNA and protein) to evaluate if the reported N protein - mediated NLRP3 inflammasome activation is transcriptional regulated or posttranslational regulated via binding of NLRP3 to ASC.

Authors' Response: Thank you for the comment.

According to your suggestion, we have carried out new experiments to determine the role of N protein in the regulation of NF- κ B signaling pathway in the revised manuscript. We noticed that the levels of NLRP3 mRNA (Revised Fig. S1c, top) as well as NLRP3, pro-IL-1 β , and pro-IL-18

proteins (Revised Fig. S1c, bottom) were relatively unaffected by N protein. These results implicated that N protein regulates NLRP3, pro-IL-1 β , and pro-IL-18 independent on NF- κ B signaling pathway. We noticed that IL-1 β protein promoted the production of inflammatory factors and cytokines, including IL-6, TNF- α , CXCL10, IL-11, IL-13, and CCL2 (Revised Fig. S1d). Thus, we speculate that N protein promotes NLRP3 inflammasome activation, thereby inducing mature IL-1 β secretion, which subsequently plays a key role in the induction of inflammatory factors and cytokines.

Reviewer #2' Comment: 5. Multiple inflammasomes uses ASC to trigger caspase-1 activation and inflammatory responses (AIM2, NLRC4, etc.). Is the N protein mediated inflammasome activation specific to NLRP3? Or N protein triggers additional ASC-dependent inflammasomes?

Authors' Response: Thank you for the suggestion.

According to your suggestion, we have explored the role of N protein in the regulation of other inflammasomes. Co-IP results showed that N protein could interact with NLRP3 protein, but failed to interact with NLRP1, NLRC4, or AIM2. Thus, we speculated that N protein triggers inflammasome activation specific to NLRP3, but not mediates ASC-dependent inflammasome activation.

Reviewer #2' Comment: Minor:

Reviewer #2' Minor Comment: 1- Does the authors have a hypothesis to explain how a protein that

is part of the envelop (therefore it is not internalized during viral entry), reaches the cytoplasm to trigger NLRP3 activation? This should be discussed.

Authors' Response: A very good suggestion.

As you suggested, we have discussed this issue in the Discussion section of the revised manuscript. Our interpretation the section of discussion is “Our results demonstrated that SARS-CoV-2 N protein, as a part of the viral envelop protein, was exited in the cytoplasm to trigger NLRP3 activation prior to viral assembly, and thereby increasing the risk of activating the acute inflammatory response.”.

Reviewer #2' Minor Comment: 2. Line 59: What the authors want so say with “the NLRP3 inflammasome is closely related to RNA virus infection”? This sentence should be adjusted.

Authors' Response: Thank you for the comment.

According to your suggestion, we have changed the description “the NLRP3 inflammasome is closely related to RNA virus infection” to “the NLRP3 inflammasome plays important role in RNA virus infection” in the revised manuscript.

Reviewer #2' Minor Comment: 3. Protein E from SARS-CoV1 is known to be porin that transports K⁺ ions and activates the NLRP3 inflammasome. Given the high homology or E from CoV-1 and Cov-2, it would be expected that E from CoV-2 triggers inflammasome. But this was not the case according to figure 1D. The authors should discuss that.

Authors' Response: Thanks again.

In our study, we noticed that like the SARS-CoV-2 N protein, the SARS-CoV-2 E protein could also induce IL-1 β production, however, the level of E-mediated IL-1 β protein was much lower than the level of N-induced IL-1 β protein (Fig. 1d, e). We speculate that SARS-CoV-2 E protein activates the NLRP3 inflammasome through a different mechanism. Further and detailed studies are needed to investigate the mechanism by which SARS-CoV-2 E protein triggers the NLRP3 inflammasome.

Reviewer #2' Minor Comment: 4. Lines 458, 460, 461, 495, 497: use the symbol “o”. It is not properly configured.

Authors' Response: Thank you for the comment.

Actually, these errors were made during the transform the PDF file from the word file.

“The lysates were rotated at 4°C for 30 min and centrifuged at 12000 rpm for 10 min to remove debris. An aliquot of supernatants was used as Input, and the rest supernatants were incubated with indicated antibodies overnight at 4°C, and mixed with Protein G sepharose beads (GE Healthcare, Milwaukee, WI, USA) for 2 h at 4°C.”.

We have made the corrections in the revised manuscript.

Reviewer #2' Minor Comment: 5. In figure 2, the authors show that N protein induced IL-1B (p17) cleavage and Casp-1 (p20) maturation in a dose dependent manner (Fig 2b, c). However, the specific concentration of N protein that induces these phenotypes in a dose dependent manner is not clear (Fig 2b)? Supplemental graphics of densitometry analysis would be useful to better visualize these data.

Authors' Response: Thank you for the comment.

According to your suggestion, we have supplemented densitometry analyses of IL-1β (p17) and Casp-1 (p20) in the revised manuscript (Revised Fig. 2b, d).

Reviewer #2' Minor Comment: 6. At Fig. 6, the legend should describe which means the solid, dashed arrows.

Authors' Response: Thank you.

Did you actually mean Fig. 8, but not Fig. 6?

There are no differences between the solid arrows and dashed arrows in original Fig. 8. To avoid the confusion, we have changed the dashed arrows to the solid arrows in the revised manuscript (Revised Fig. 8).

Reviewer #2' Minor Comment: 7. IL-18 and Inflammasome activation have been identified as biomarkers for mortality after SARS CoV-2 infection in patients (Rodrigues et al., JEM 2020; Lucas et al., Nature 2020). However, in the present study, the authors did not observe N protein effects on the IL-18 expression in mouse. Did the authors measure IL-18 expression in the mice sera? The authors should at least discuss these findings in the light of previous publications.

Authors' Response: Thank you for the comment.

According to your suggestion, we have performed additional experiments supplemented the IL-18 expression of the mice serum in the revised new animal experiments. The results showed that: compared with AAV-N infected NLRP3^{+/+} groups. The levels of IL-18 in mice serum (Revised Fig. S6c, d) were similar with both in pre-treated with MCC950 groups and NLRP3^{-/-} groups. IL-18 have been identified as biomarkers for mortality after SARS CoV-2 infection in patients (Rodrigues et al., JEM 2020; Lucas et al., Nature 2020), but IL-18 maybe not to be regulated by the N protein.

REVIEWER COMMENTS

Reviewer #1 (Remarks to the Author):

The authors have addressed my initial concerns with the publication with the addition of substantial new data and the inclusion of suitable controls.

I have two minor questions for the authors which they may wish to address in their manuscript. Figure S2 demonstrates the induction of IL-1b by N protein, and have now included ORF3a as a control- however, the authors results suggest that ORF3a does indeed induce a significant increase in IL-1b secretion, yet in the text (line 177) the authors state the IL-1b levels are not affected by ORF3a. This significant increase according to the authors analysis needs to be addressed or clarified.

In the rebuttal the authors content that ORF3a does not form specks or spheres in cells compared to N protein. However, Fig 6j shows clear 3a-induced specks in A549 cells in panels ae-ai- particularly panel af. Conversely, figure 6a does not clearly demonstrate N protein-induced specks in A549 cells (especially compared to HEK293T cells), whereas the argument could be made for specks in the lower left quadrant of the 3a/NLRP3 cell shown. Could the authors please clarify.

Reviewer #2 (Remarks to the Author):

The authors addressed my comments but a few questions remain:

In relation to my comment #2, the authors performed experiments using MCC950 as well as the NLRP3 KO mice.

The authors performed experiments using MCC950 in mouse and using NLRP3 KO mice. This is fine, but using the in vivo approach they inhibit a systemic effect of NLRP3 inflammasome. This is evident in the revised figure 1, where they show MCC950 promotes general reduction in gene expression (Fig. 1g and 1i). MCC950-mediated inhibition of inflammasome activation should be independent of transcription regulation of inflammatory cytokines. So, the authors are probably reporting a feedback loop downstream of NLRP3 inflammasome activation (possibly via IL-1 β) that led to transcriptional regulation of inflammatory genes.

To support the author's claim that N protein triggers NLRP3 directly, they should test in vitro. I think it is important to performed experiments such as those in Fig. 2 and Fig. 6 using MCC950 (in case of Fig. 2a, c, d...) and NLRP3 KO (in case of Fig. 3g).

In relation to my question #4: "N protein triggers additional ASC-dependent inflammasomes?"

The authors tested if N protein Co-IP with NLRP3, NLRP1, NLRC4, or AIM2. This experiment is OK and should be part of the manuscript (and not only for the reviewer's evaluation). Importantly, this experiment tests the binding and not activation. They should test if the N-mediated inflammasome activation (observed in the mock of Fig. 1f for example) is reduced in absence of NLRP3. They should also induce AIM2, NLRP1 or NLRC4 inflammasomes and see if they find the same result observed for Nigericin or LPS+ATP (in Fig. 2).

These experiments are particularly important because the authors claim this is a NLRP3 specific phenomenon. This statement is in all over the manuscript, including in the title.

In the light of the data presented, it is still possible that the reported phenomenon is not mediated by a direct effect of N protein in NLRP3. Alternatively, it could be a general nonspecific proinflammatory effect of N protein, which is amplified by the NLRP3 inflammasome.

Response to the Reviewers

REVIEWER COMMENTS

Reviewer #1 (Remarks to the Author):

The authors have addressed my initial concerns with the publication with the addition of substantial new data and the inclusion of suitable controls.

Reviewer

Authors' Response: Thank you very much for your remarks.

I have two minor questions for the authors which they may wish to address in their manuscript.

Figure S2 demonstrates the induction of IL-1b by N protein, and have now included ORF3a as a control- however, the authors results suggest that ORF3a does indeed induce a significant increase in IL-1b secretion, yet in the text (line 177) the authors state the IL-1b levels are not affected by ORF3a. This significant increase according to the authors analysis needs to be addressed or clarified.

Authors' Response: Thank you for the good comment.

As you suggested, to clarify this result we have changed the statement “Interestingly, the level of IL-1 β protein was significantly induced by the SARS-CoV-2 N protein, but not affected by the SARS-CoV-2 3a protein (Fig. S2), confirming a specific role of N protein in the activation of IL-1 β protein” to “Interestingly, the level of IL-1 β protein was significantly induced by the SARS-CoV-2 N protein, and slightly enhanced by the SARS-CoV-2 3a protein (Fig. S2), suggesting a specific role of N protein in the activation of IL-1 β protein” in the revised manuscript (Revised lines 177-178).

In the rebuttal the authors content that ORF3a does not form specks or spheres in cells compared to N protein. However, Fig 6j shows clear 3a-induced specks in A549 cells

in panels **ae-ai**-particularly panel **af**. Conversely, **figure 6a** does not clearly demonstrate N protein-induced specks in A549 cells (especially compared to HEK293T cells), whereas the argument could be made for specks in the lower left quadrant of the **3a/NLRP3** cell shown. Could the authors please clarify.

Authors' Response: Thanks for the comment.

Sorry for the confusion. Actually, in A549 cells transfected with plasmids expressing GFP, NLRP3, and ASC, the two proteins (NLRP3 and ASC) could form “speck” structures, as expected, even in the absence of N protein or 3a protein (**Revised Fig. 6j u-y**). Importantly, we noticed that when N, ASC, and NLRP3 expressed together, the three protein were obviously co-localized to form “sphere-like” structures (**Revised Fig. 6i z-ad, Fig. 6j z-ad, Movie S4, and Movie S5**), in which N was in the center, NLRP3 was in the middle, and ASC was in the outside. Notably, we also noticed that when 3a, ASC, and NLRP3 expressed together, although they could form “speck” structures, but failed to form “sphere-like” structures (**Fig. 6i ae-ai and Fig. 6j ae-ai**). Therefore, these data suggest that N protein, but not 3a protein, is involved in the facilitation of the assemble of the NLRP3 inflammasome.

To make it clearer, we have repeated the experiments in A549 cells in the revised manuscript. The new results clearly showed that N protein and NLRP3 protein could co-localize and form specks (**Revised Fig 6a bottom, middle panels**). However, in the presence of SARS-CoV-2 3a protein, NLRP3 was co-localized with 3a, but failed to form specks (**Revised Fig 6a bottom, lower panels**). These results also indicate that N protein, but not 3a protein, facilitates the NLRP3 inflammasome assemble.

Reviewer #2 (Remarks to the Author):

The authors addressed my comments but a few questions remain:

In relation to my comment #2, the authors performed experiments using MCC950 as well as the NLRP3 KO mice.

Authors' Response: Thank you very much for your remarks.

The authors performed experiments using MCC950 in mouse and using NLRP3 KO mice. This is fine, but using the in vivo approach they inhibit a systemic effect of NLRP3 inflammasome. This is evident in the revised figure 1, where they show MCC950 promotes general reduction in gene expression (Fig. 1g and 1i). MCC950-mediated inhibition of inflammasome activation should be independent of transcription regulation of inflammatory cytokines. So, the authors are probably reporting a feedback loop downstream of NLRP3 inflammasome activation (possibly via IL-1 β) that led to transcriptional regulation of inflammatory genes. To support the author's claim that N protein triggers NLRP3 directly, they should test in vitro. I think it is important to performed experiments such as those in Fig. 2 and Fig. 6 using MCC950 (in case of Fig. 2a, c, d...) and NLRP3 KO (in case of Fig. 3g).

Authors' Response: Thank you for the comments.

According to your suggestion, we have performed additional experiments to support our claim that N protein triggers NLRP3 directly in the revised manuscript.

We have further investigated the role of NLRP3 involved in IL-1 β maturation and IL-6 production induced by the N protein. THP-1 cells stably infected with CT-Lentivirus and N-Lentivirus were differentiated into macrophages, which were pre-treated with MCC950 (a specific inhibitor of the NLRP3) and then stimulated with LPS plus ATP or LPS plus Nigericin. Notably, in the presence of LPS plus ATP or LPS plus Nigericin, secreted IL-1 β protein was significantly induced by N protein, but N-induced secreted IL-1 β protein was remarkably suppressed by MCC950 (Revised Fig. 2m). Similarly, IL-6 protein was enhanced by N protein, while N-enhanced IL-6 protein was also repressed by MCC950 (Revised Fig. 2n). These results suggest that N protein induces IL-1 β secretion and IL-6 production depending on NLRP3. Moreover, GM-CSF differentiated BMDMs of wide-type mice and NLRP3^{-/-} mice were infected with N-Lentivirus and then stimulated with LPS, LPS plus ATP, and LPS plus Nigericin. We noticed that the levels of secreted IL-1 β protein (Revised Fig. 2o) and IL-6 protein (Revised Fig. 2p) induced by N protein were significantly lower in the BMDMs of NLRP3^{-/-} mice compared with that in BMDMs of wide-type mice (Revised Fig. 2o, p). These data indicate that N protein fails to

induce IL-1 β protein and IL-6 protein in NLRP3^{-/-} mice BMDMs, and thereby suggesting that NLRP3 is required for IL-1 β secretion and IL-6 production induced by the N protein. Taken together, our results demonstrate that N protein induces IL-1 β maturation and IL-6 production, reveal that NLRP3 is required for N-induced IL-1 β secretion and IL-6 production protein, and thereby suggesting that N protein may play an important role in the activating the NLRP3 inflammasome.

In addition, we showed that SARS-CoV-2 N protein significantly induced ASC oligomerization, but SARS-CoV-2 3a protein had no effect on ASC oligomerization (Revised Fig. S6b). Interestingly, in PMA-differentiated THP-1 macrophages (Revised Fig. 6f) and GM-CSF differentiated BMDMs (Revised Fig. 6g), oligomerization of endogenous ASC protein was stimulated by Nigericin, and Nigericin-induced ASC oligomerization was further enhanced in the presence of N protein (Revised Fig. 6f, g). However, N protein induced endogenous ASC oligomerization was significantly inhibited by MCC950 in PMA-differentiated THP-1 macrophages (Revised Fig. S6c) and GM-CSF differentiated NLRP3^{-/-} BMDMs (Revised Fig. S6d). Taken together, these in vitro experiments along with in vivo experiments (Revised Fig. 7 and Revised Fig. S8) demonstrate that N protein triggers NLRP3 directly.

In relation to my question #4: “N protein triggers additional ASC-dependent inflammasomes?”

The authors tested if N protein Co-IP with NLRP3, NLRP1, NLRC4, or AIM2. This experiment is OK and should be part of the manuscript (and not only for the reviewer’s evaluation). Importantly, this experiment tests the binding and not activation. They should test if the N-mediated inflammasome activation (observed in the mock of Fig. 1f for example) is reduced in absence of NLRP3. They should also induce AIM2, NLRP1 or NLRC4 inflammasomes and see if they find the same result observed for Nigericin or LPS+ATP (in Fig. 2).

Authors’ Response: Thank you for your comments and suggestions.

According to your suggestion, we have added the result in revised manuscript (Revised Fig. S4a). Co-immunoprecipitation (Co-IP) assays showed that N protein only interacted with the NLRP3 protein and failed to interact with the NLRP1, NLRC4, or AIM2 proteins (Revised Fig. S4a).

As you suggested, in addition, we have performed additional experiments in the revised manuscript. Notably, the new ELISA results showed that N protein could induce IL-1 β production in the presence of NLRP3, but failed to promote IL-1 β production in the presence of NLRP1, NLRC4, or AIM2 (Fig. S4b). Similarly, our new data indicated N protein specifically induced NLRP3 inflammasome regulated ASC oligomerization, but failed to induce NLRC4 inflammasome or AIM2 inflammasome regulated ASC oligomerization (Fig. S6a).

Interestingly, in PMA-differentiated THP-1 macrophages (Revised Fig. 6f) and GM-CSF differentiated BMDMs (Revised Fig. 6g), oligomerization of endogenous ASC protein was stimulated by Nigericin, and Nigericin-induced ASC oligomerization was further enhanced in the presence of N protein (Revised Fig. 6f, g). However, N protein induced endogenous ASC oligomerization was significantly inhibited by MCC950 in PMA-differentiated THP-1 macrophages (Revised Fig. S6c) and GM-CSF differentiated NLRP3^{-/-} BMDMs (Revised Fig. S6d). Taken together, these results demonstrate that N protein specifically and directly triggers NLRP3.

These experiments are particularly important because the authors claim this is a NLRP3 specific phenomenon. This statement is in all over the manuscript, including in the title.

In the light of the data presented, it is still possible that the reported phenomenon is not mediated by a direct effect of N protein in NLRP3. Alternatively, it could be a general nonspecific proinflammatory effect of N protein, which is amplified by the NLRP3 inflammasome.

Authors' Response: Again, thank you for the comments.

To address your comments and according to your suggestions, we have performed

additional experiments in the revised manuscript. The new data (Revised Fig. 2m–p, Revised Fig. S4a–b, and Revised Fig. S6a–d) along with our previous results all demonstrate that N protein triggers NLRP3 directly, rather than just amplifying the NLRP3 inflammasome.

REVIEWERS' COMMENTS

Reviewer #2 (Remarks to the Author):

The authors have addressed all my comments.

Response to the Reviewers

REVIEWERS' COMMENTS

Reviewer #2 (Remarks to the Author)

The authors have addressed all my comments.

Authors' Response: Thank you very much for your comment.